# CreDes: Causal Reasoning Enhancement and Dual-End Searching for Solving Long-Range Reasoning Problems using LLMs

## Abstract

Large language models (LLMs) have demonstrated limitations in handling combinatorial optimization problems involving long-range reasoning, partially due to causal hallucinations and huge search space. As for causal hallucinations, i.e., the inconsistency between reasoning and corresponding state transition, this paper introduces the Causal Relationship Enhancement (CRE) mechanism combining cause-effect interventions and the Average Treatment Effect (ATE) to guarantee the solid causal rightness between each step of reasoning and state transition. As for the long causal range and huge search space limiting the performances of existing models featuring single-direction search, a Dual-End Searching (DES) approach is proposed to seek solutions by simultaneously starting from both the initial and goal states on the causal probability tree. By integrating CRE and DES (CreDes), our model has realized simultaneous multi-step reasoning, circumventing the inefficiencies from cascading multiple one-step reasoning like the Chain-of-Thought (CoT). Experiments demonstrate that CreDes significantly outperforms existing State-Of-The-Art (SOTA) solutions in long-range reasoning tasks in terms of both accuracy and time efficiency.

## 1 Introduction

Reasoning aims to realize the causal transfer from the initial state to the goal state through several intermediate steps, which widely exists in the domains of Societal Simulation[1, 2, 3], Economic Simulation[4, 5, 6], Game Theory[7, 8, 9] and Gaming[10, 11, 12], etc. LLMs like GPT-3 have shown competitive performances in many reasoning tasks[13, 14, 15]. However, their performances and efficiency are limited when dealing with complex combinatorial optimization problems that require multi-step long-range reasoning[16].

The first challenge is causal hallucinations, i.e., causality between one-step reasoning (OSR) and state transition in LLMs is not always guaranteed. Similar to pre-trained LLMs that are prone to produce hallucinations when processing certain factual information, causal hallucinations reflect the fact that LLMs lack rigor due to inherent randomness in accomplishing complex mathematical[17, 18, 19], logical[20, 21], or common-sense reasoning[22, 23, 24], which is somehow entrenched in statistical inevitability and independent of the Transformer architecture or data quality[25]. For example, CoT-based finite-step reasoning methods[26, 27] suffer from causal hallucinations, which cannot effectively ensure the causality between OSR and state transition in LLMs, resulting in unreliable reasoning and relatively low success rates (especially for long-range reasoning problems with significant error accumulation effects). The reasonableness between OSR and state transition can be summarized as follows: There is a causal relationship between reasonable OSR and state transition. At the same time, there is only a correlation or no relationship between unreasonable OSR and state transition, which

suggests that training with cross-entropy loss alone does not enable the model to have sufficient causal rigor. Inspired by this, we designed the CRE mechanism to make each step of reasoning correct and *causally sound* by including the causality measure between OSR and state transition as part of the training objective, thus more closely modeling the rigor, adaptability, and comprehensiveness of human reasoning[28].

The second challenge is that long-range reasoning problems have a huge search space. Although complex architectures such as CoT, Tree of Thought (ToT)[29], and Program of Thought (PoT)[30] can effectively improve the reasoning accuracy of LLMs through external guidance, they are limited when handling long-range reasoning processes and task decomposition. A crucial reason is that long-range reasoning has a huge state space, i.e., each branch in the state transition process expands the search space approximately exponentially. Most of the existing LLM-based methods, e.g., Monte Carlo search[31], are based on unidirectional reasoning, making them time-inefficient and easy to fall into local optima when dealing with reasoning problems with large search spaces. In this paper, a bi-directional Dual-End Searching method is developed, which first decomposes a long-range reasoning problem into a combination of short-range reasoning problems and then searches for the intersection of two causal probability trees starting from the initial and goal states, respectively.

A structured and generalized reasoning framework, CreDes, is developed for long-range reasoning with LLMs in this paper, and the contributions can be summarized as follows:

**First, the CRE mechanism is introduced to improve the rigor of LLM-based long-range reasoning methods:** Structural Causal Modeling (SCM) is exploited to enhance the causality between OSR and state transitions, involving performing causal interventions and optimizing the absolute value of ATE during training, which has effectively alleviated causal hallucinations in long-range reasoning of LLMs.

**Second, the DES method is developed to improve the search efficiency for long-range reasoning:** After constructing causal probability trees starting from the initial states and ending at the goal states, long-range reasoning (e.g., 12 steps) is transformed into more manageable combinations of smaller segments (e.g., 2 or 4 steps) by minimizing the distances between leaves of the tree and employing end-matching techniques. By avoiding long-range sequential search from scratch, the DES method has greatly lowered the complexity when solving long-range reasoning problems.

**Third, simultaneous multi-step reasoning is realized to improve the time-efficiency of long-range reasoning:** By integrating CRE and DES, CreDes can perform simultaneous multi-step reasoning within the model, i.e., avoiding the inefficiency of *cascading single-step reasoning* in frameworks such as CoT. While ensuring the accuracy of the reasoning process, CreDes can significantly reduce the time required for multi-step reasoning in LLMs.

**Fourth, adequate and rigorous testing of CreDes:** CreDes has been extensively tested in the Blocksworld, GSM8K, and Hanoi Tower scenarios, respectively, and the experimental results show that CreDes outperforms existing SOTA regarding reasoning accuracy and time efficiency.

## 2   Related Work

**Decision-Making Capabilities in LLMs:** The core of intelligence partially lies in planning, which encompasses generating a sequence of actions aimed at accomplishing a predefined objective[32, 33]. Classical planning methods have found extensive application in robotics and embodied environments, where they are commonly employed to guide decision-making processes externally[34, 35]. Recent advancements, such as the Chain-of-Thought model[26, 36, 37], have significantly bolstered the LLMs' capability to perform detailed reasoning[38, 39, 40]. This model breaks down intricate queries into a series of manageable steps, thereby enhancing the LLMs' decision-making ability. Subsequent initiatives like ReACT[41] have modified this approach to improve reasoning ability in decision contexts using a CoT-based framework. Additionally, Reflexion[42] provides a corrective mechanism that enables LLMs to recognize their errors during the decision-making process, reflect on these mistakes, and make accurate decisions in subsequent attempts. Further developments have led to the creation of tree-based decision-making frameworks that tailor LLM capabilities to specific scenarios. The Tree-of-Thought[29] utilizes Breadth First Search (BFS) and Depth First Search (DFS) algorithms to facilitate decision-making in activities such as the Game of 24, Creative Writing, and Mini Crosswords. Meanwhile, Reasoning via Planning (RAP)[43] employs the Monte Carlo

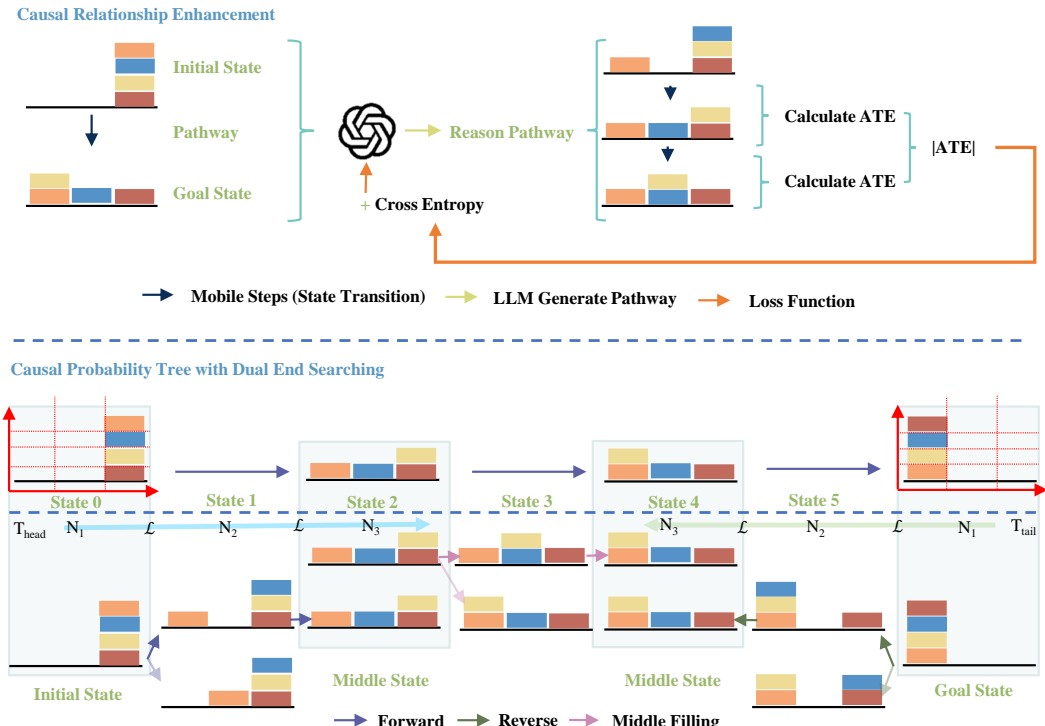

Figure 1: Integrating Causal Relationship Enhancement (CRE) and Dual-End Searching (DES).

Tree Search technique to optimize solutions across tasks like Blocksworld[44], Math Reasoning[45]. DFSDT[46] proposed an efficient version of DFS for LLMs to make decisions, but it lacks the judgment ability to evaluate different decisions. JUDEC[47] utilizes an Elo rating system to enable LLMs to develop self-assessment capabilities, thereby enabling them to generate optimal solutions for a wide range of real-world tasks, independent of any task-specific expertise. Lastly, Graph-of-Thought[48] represents the thoughts as nodes in a graph, combining thoughts non-sequentially. Encouraged by the studies above, we leveraged LLMs to solve long-range reasoning problems.

**Integrating Causal Analysis in LLMs for Multi-step Decision-Making:** The causal analysis aims to discern and elucidate the causal relationships between actions, circumstances, or decisions. This method entails investigating the origins or causes leading to an event and the potential consequences that follow[49, 50, 51]. Although various causal models may produce identical observational distributions, they can yield distinct distributions when interventions are applied[52]. Therefore, using interventions allows for the distinction of possible causal frameworks that align with the observed data[53, 54]. Previous work suggests that, while CoT has been lauded for its potential to improve task performance, its application does not always lead to enhanced outcomes[36, 55]. Also, research has shown that the statistical pretraining of LLMs encourages models to achieve high empirical performance but not necessarily to reason[56, 57, 58, 59]. Inspired by this, we designed the CRE mechanism to control the causal hallucinations of LLMs to solve long-range reasoning problems.

**Solving Multi-step Problems with LLMs:** Recent studies have shown that with substantial design, LLMs are capable of performing not only basic arithmetic tasks but also complex multi-step reasoning[60, 61]. For instance, increasing computational resources significantly enhances the accuracy of datasets like GSM8K[62]. Concurrently, Research[63] demonstrated that a 2B parameter LLM could achieve 89.9% accuracy in 5x5 multiplication tasks using curriculum learning with 50 million training instances. This evidence suggests that adequately scaled LLMs can process multiple reasoning steps effectively internally. While trees are frequently used to represent games (especially extensive-form games[64, 65]) and sequential reasoning problems[66], it was Shafer's groundbreaking work[67] that initially established a framework for understanding causality through the use of probability trees. Inspired by Shafer's approach, we recognized that LLMs tend to struggle with long-range reasoning problems involving multiple steps but excel in short-range reasoning tasks.

By integrating causal probability trees, we can enhance search efficiency. This insight led to the development of DES.

# 3 Method

The pipeline of CreDes is illustrated in Fig. 1. It comprises two main components: CRE and DES. In CRE, the inputs of LLMs for training are the initial state, goal state, and pathway (containing a series of OSRs), while for testing, the inputs are the initial and goal states only. The DES starts from the initial and goal states of the probability tree, expands them into two intermediate states, and uses the CRE-trained model to infer the pathway between them, ultimately producing the complete pathway.

## 3.1 Problem Definition

To further improve the capability of LLMs in solving combinatorial optimization problems that involve a finite number of discrete intermediate steps, we conducted experiments using the Blocksworld and Hanoi Tower datasets with 7B parameter models. The Blocksworld dataset includes 602 test cases categorized by the minimum number of required actions, ranging from 2 to 12 steps. For Hanoi Tower, cases are grouped based on the complexity related to the number of disks and poles, which directly influences the solution steps.

For each category, our model is trained on 80 samples without common instructions. In the reasoning process, the following elements are included: initial state, OSR, state transition, next state, and goal state, as shown in Fig. 2. During testing, the model was tested on new, categorically similar samples from different datasets, assessing its ability to transform the initial state to the goal state successfully.

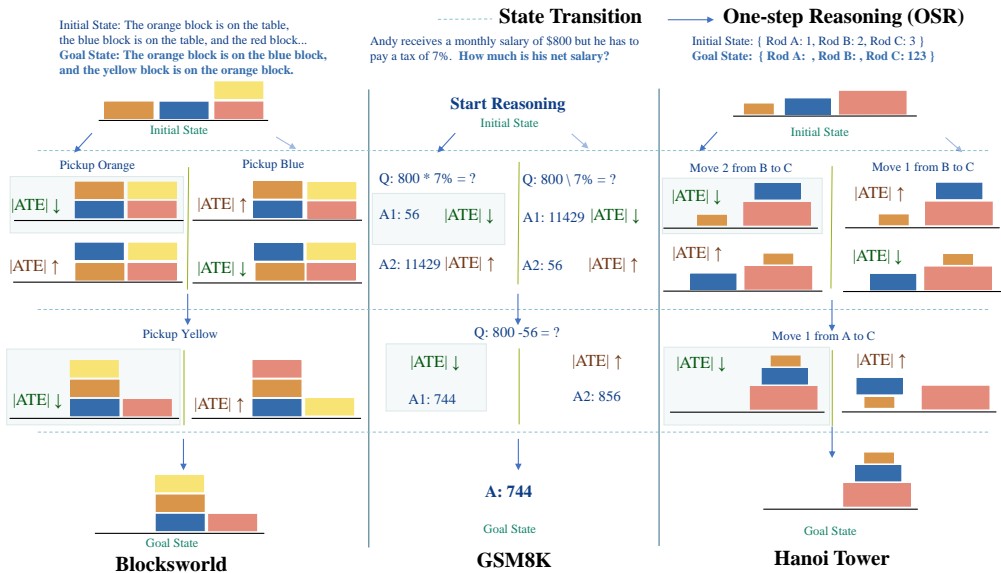

Figure 2: Schematic illustration of Causal Relationship Enhancement(CRE).

## 3.2 Causal Relationship Enhancement (CRE)

Firstly, all the samples are classified into two categories: Correct and Incorrect. Within the Incorrect category, three scenarios exist, i.e., a correct OSR leading to an incorrect state transition, an incorrect OSR leading to an incorrect state transition, and an incorrect OSR resulting in a correct state transition. Given this, it is evident that we need to strengthen the causal connection between the OSR and the transition, and reduce the occurrence of samples where the OSR and the transition are non-causal. In CRE, we first use the ATE to estimate the causality between OSR and state transition quantitatively, and then embed the |ATE| into the loss function in the training process (the remaining is cross-entropy), enhancing the causality of state transitions. As is shown in Fig. 2, we leave the reasoning

path selection to be controlled by the cross-entropy loss, while the suppression of hallucinations is handled by the |ATE| loss. Perplexity (PPL) is a metric used to evaluate the performance of a LLM, indicating how well the model predicts the next word in a sequence, and lower values signify better predictive accuracy. The estimation of ATE is detailed as the follows:

Given binary variables $X$ and $Y$ indicating the correctness of OSR and next state (state transition), respectively, i.e., $X, Y \sim B(0,1)$, and $X = 1$ (or $Y = 1$) means correctness. First, we calculate the cause-effect interventions between $X$ and $Y$, then subsequently modify the distribution of $Y$ by intervening in $X$. From a statistical correlation perspective, if $X$ and $Y$ are correlated, $Y$ can be predicted using $X$. However, if there is no causal relationship between $X$ and $Y$, intervening in $X$ will not alter the distribution of $Y$. Hence, if $X$ and $Y$ are correlated but not causally linked, then manipulating or intervening in $X$ would not lead to any changes in the distribution of $Y$. This distinction is crucial in statistical analysis and experimental design because it addresses the potential fallacy that correlation inherently means causation.

$$P_Y^{do(X)} = P(Y|do(X = 1)) - P(Y|do(X = 0)) \tag{1}$$

In (1) and (2), $do(\cdot)$ refers to Do-calculus[68], which denotes an external intervention on the value of $X$ without affecting the actual state of $Y$. Using interventions independent from other variables, we can obtain whether the treated variable $X$ causes the target variable $Y$. Consequently, we can use ATE[69] to estimate the effect of the intervention, which compares the distributions of the target variable $Y$ with and without the treatment. $Y_x$ is the potential outcome of $Y$ under the intervention $X = x$. Then ATE is defined as follows:

$$\text{ATE} = E(Y|do(X)) - E(Y) = E[Y_1 - Y_0] \tag{2}$$

Based on (2), under the intervention, the proportion of positive and negative cases (hallucinations) in the model output samples remains roughly unchanged; the more robust the causal relationship between different OSRs and corresponding positive and negative cases, the lower the |ATE|. The reason is that cross-entropy basically ensures the majority of positive cases. At the same time, |ATE| reduces the occurrence of negative cases, making the distribution of positive and negative cases more stable. Consequently, we incorporate the ATE into the loss function, as is shown in (3) and (4), $p_{1|X}$ and $p_{0|X}$ denote the conditional probabilities of $Y$ being *1* and *0*, respectively, given the state of $X$.

$$\mathcal{L}_{CrossEntropyLoss} = - \left[ Y \log(p_{1|X}) + (1 - Y) \log(p_{0|X}) \right] \tag{3}$$

$$\mathcal{L}_{Loss} = \mathcal{L}_{CrossEntropyLoss} + |\text{ATE}| = \ln(\text{PPL}) \tag{4}$$

We estimated the probabilities of correct and incorrect (hallucinations) samples in each path separately, and take |ATE| as part of the loss function for each case based on the sampling results between different paths, where |ATE| is smaller for the category with strong causal effects. This process allows the model to internalize the logical judgment between OSR and the next state during training, i.e., correct answers with strong causal effects and low |ATE|, and wrong answers with weak causal effects and high |ATE|. Therefore, with the loss function composed of cross-entropy and ATE, we can realize the synergistic optimization of path selection and hallucination elimination simultaneously.

### 3.3 Causal Probability Trees with Dual End Searching (DES)

In this section, we improve the success rate of LLMs when solving long-range reasoning problems, like the 12-step scenarios in Blocksworld, by leveraging its higher success rates in simpler 2-step and 4-step scenarios. We construct two causal probability trees from the initial and goal states. Each node represents a state in the reasoning process, with arrows showing causal relationships. These trees outline possible reasoning outcomes within a limited number of intermediate steps. By matching the leaves of both trees, we identify several end-to-end permutation schemes to form a continuous and feasible path, as shown in the bottom of Fig. 1.

DES first calculates the ATE between tree unfolding and distance reduction, which in turn clarifies the causal relationship between tree unfolding and distance reduction and then infers a better unfolding direction and pruning process. At this point, the ATE calculation formula is:

$$\text{ATE}(A) = E(A|do(B)) - E(A) \tag{5}$$

Where $A$ is the decrease in distance $D$ of $N_i$ relative to $N_{i-1}$ and $B$ is the number of unfolded layers where the current leaf is located $N_i$. We utilize the spatial positioning numbers of the blocks

(disks) to calculate the distance $D$, and estimate the ATE of the reduction in distance, denoted as $\delta D$, relative to the number of layers $N_i$ of tree expansion. The distance is obtained by calculating the Euclidean distance for the current position of the block and the coordinates of the target position.

In each layer of the tree expansion, we calculate the distance by comparing the current position with their target positions, ensuring that the reduction in distance and the direction of the tree's expansion have a strong causal effect. To avoid the expansion direction falling into local optimum, we conduct counterfactual assessments, hypothesizing alternative expansion routes that might have been taken during the random expansion process, and incorporating the causal impacts of these hypothetical routes into consideration. Both these values are summed up to form the loss function, taking into account both the head $T_{head}$ and tail $T_{tail}$ trees.

$$\mathcal{L} = |\text{ATE}(\delta D_{T_{head}}^{N_i - N_{i-1}})| + |\text{ATE}(\delta D_{T_{tail}}^{N_i - N_{i-1}})| + D \tag{6}$$

During the expansion process of the probability trees at both ends, we intervene by minimally altering the $\mathcal{L}$, directing the expansion toward our desired outcome. Minimizing $\mathcal{L}$ realizes the pruning and unfolding direction judgment, prioritizing the direction with the lowest $\mathcal{L}$ as the unfolding direction. The whole process of DES is in **Algorithm 1**.

---

**Algorithm 1** DES (Taking the 12-step Blocksworld as an example)

---

1: **Input:** $State_{init}$ and $State_{goal}$, denoting the initial and goal states, respectively
2: **Output:** Complete 12-step solution process
3: Construct $T_{head}$ and $T_{tail}$ from $State_{init}$ and $State_{goal}$
4: Match leaves of $T_{head}$ and $T_{tail}$ to form paths
5: **for** every four steps **do**
6:     Determine intermediate steps and fill in details
7: **end for**
8: **for** expanding $T_{head}$ and $T_{tail}$ **do**
9:     Calculate distance $D$
10:     Minimize $\mathcal{L}$
11:     **if** local optimum detected **then**
12:         Assess alternative routes
13:     **end if**
14: **end for**

---

# 4 Experiment

In this section, we validated the effectiveness of CreDes compared to baseline approaches.

## 4.1 Setup

**Blocksworld:** There are $n$ blocks initially placed randomly on a table[44]. The LLM's goal is to stack these blocks in a specified order. The LLM can perform four actions: pick up a block from the table, put down a block it is holding onto the table, unstack a block from another to hold it, and stack the block in its hand onto another block. The LLM can only manipulate one block at a time, and blocks with others on top are immovable.

**GSM8K:** The GSM8K dataset[62] includes 1,319 diverse grade school math word problems curated by human problem writers. These tasks typically begin with a description and culminate in a final question requiring multi-step mathematical calculations contextual to the problem. To effectively tackle the final question, our approach involves decomposing it into a sequential series of smaller sub-questions, allowing for a structured solution process.

**Hanoi Tower:** The Hanoi Tower problem[70], a classic puzzle involving three pegs and a set of discs of varying sizes, serves as a key component of our experimental setup. The challenge requires moving the entire stack of discs from one peg to another, obeying the rules that only one disc can be moved at a time, and no disc may be placed on top of a smaller one. This task, structured around sequential and strategic disc placement, tests the model's ability to plan and execute a series of actions based on simple yet strict rules.

## 4.2 Dataset and Basemodel

**Dataset:** The datasets we used are the open source datasets Blocksworld[44], GSM8K[62], AQUA[71], QASC[72], and our own production of Hanoi Tower. where the experiments for AQUA and QASC are in the Table 4.

**Basemodel:** The pre-trained models used in our study include: LLAMA-2-7B[73], Phi-2-7B[74], Mistral-7B[75] and Mixtral-8x7B[76], Qwen1.5-7B[[77]], TAIDE-LX-7B[1], Mpt-7B[[78]], Baichuan2-7B[[79]],The model test results not mentioned in the main text will be supplemented in the Appendix.

## 4.3 Benchmark

**Train Parameter:** In this paper, we primarily utilize the 7B models for training on a single NVIDIA A100 GPU and models are loaded in 4-bit.

**RAP:** A technique that employs Monte Carlo Tree Search (MCTS) for exploration[43]. RAP transforms LLMs into both reasoning agents and world models, utilizing MCTS for strategic exploration and decision-making. This approach significantly enhances the LLM's ability to generate action plans and solve mathematical and logical problems, outperforming traditional methods and establishing new benchmarks in LLM's capabilities.

**CoT[26]:** A technique having enhanced the reasoning capabilities of LLMs. By providing models with intermediate reasoning steps as examples, CoT demonstrates notable improvements across various complex reasoning tasks, including arithmetic, commonsense, and symbolic reasoning. CoT requires the model to generate a reasoning chain to improve the reasoning ability. We used all basemodels to carry out CoT in the experiment.

**RoT:** A framework[80] to enhance the performance of tree-search-based prompting methods used in LLMs. This innovative approach leverages guidelines derived from past tree search experiences, allowing LLMs to avoid repeating errors and significantly improving their reasoning and planning capabilities across various tasks. We not only used the same basemodel as the original RoT, but also introduced other 7B models as a comparison.

## 4.4 Results

**Blocksworld:** We conducted ablation experiments on the Blocksworld dataset. Our methodology, detailed in Section 3, particularly focuses on scenarios with more than 6 steps. As is shown in Table 1 and Table 5, for tasks up to 6 steps, results with our 7B models closely matched those with the benchmark's 70B models, suggesting robust inference capabilities even with reduced model size. For more complex tasks of 8 steps or more, DES improved its success rates by breaking down tasks into simpler segments, though it slightly lagged behind in performance compared to shorter tasks. This approach underlines the potential of our modified strategies in handling varying task complexities. By comparison, our CRE method not only outperforms benchmarks in terms of success rates on the 7B scale, but also achieves a higher success rate than the 70B+RAP method using the 7B model. For the arithmetic cases that use the full CreDes architecture, CreDes helps to improve the performance of the LLMs for long-range reasoning tasks.

**GSM8K:** We further independently verified the capabilities of CRE based on the GSM8K dataset without introducing DES, to confirm that it helps to enhance the inference capabilities of large models. We found that our CRE is superior to the baseline methods RAP, RoT, and CoT, further demonstrating that completing multi-step reasoning in one go has more advantages than completing multiple single-step reasoning. See Table 2. This example shows that CRE can not only help LLM solve highly structured problems, such as Blocksworld, but also has the ability to assist in solving some abstract mathematical problems.

**Hanoi Tower:** Unlike the Blocksworld case, the longest reasoning steps for the Hanoi Tower have a fixed quantitative relationship with the number of rods and disks. Therefore, when training the model, we used combinations within 7 steps, i.e., 3 rods and 3 disks. For evaluation, we used problems within 15 steps, i.e., combinations of 3 rods and 4 disks, to test the reasoning ability. From this

---

[1]http://taide.tw

perspective, our reasoning process is based on a zero-shot setting. Due to the time complexity of the search-based method for long-range reasoning, we did not conduct experiments for too many reasoning steps, and its success rate can be recorded as '-.' As Table 3 shows, CreDes performed best among all the models. By comparing the Hanoi Tower scenario with the Blocksworld scenario, we find that the success rate under Hanoi Tower is lower than that of Blocksworld, and that the reasoning ability of the 7B+CRE group is slightly lower than that of the 70B+RAP group. We believe that this phenomenon occurs because Hanoi Tower has a stricter stacking order qualification relative to Blocksworld, and some of the intermediate steps may not hold at all, see Fig. 2. From the results, the complexity of the Hanoi Tower problem is higher than that of Blocksworld.

**Time Efficiency:** Using the CRE and DES architecture has significantly shortened the time to complete long-range reasoning tasks compared to benchmarks, as is shown in Fig.3. This is because CreDes can perform simultaneous multi-step reasoning, which is more efficient than other methods that generate answers multiple times and then cascade them together, which is more evident in longer-range reasoning.

Table 1: Succcess Rate under Blocksworld

| Model | 2-step | 4-step | 6-step | 8-step | 10-step | 12-step |
|---|---|---|---|---|---|---|
| Llama-2-70B + RAP | 0.67 | 0.76 | 0.74 | 0.48 | 0.17 | 0.09 |
| Llama-2-7B + RAP | 0.39 | 0.41 | 0.37 | 0.11 | 0.00 | 0.00 |
| Llama-2-7B + CoT | 0.50 | 0.63 | 0.40 | 0.27 | 0.07 | 0.00 |
| Llama-2-7B + RoT | 0.52 | 0.67 | 0.27 | 0.06 | 0.00 | 0.00 |
| Llama-2-7B + CRE | **0.95** | **0.80** | **0.76** | 0.22 | 0.09 | 0.00 |
| Llama-2-7B + CreDes | - | - | - | **0.68** | **0.51** | **0.34** |
| Phi-2-7B + RAP | 0.40 | 0.44 | 0.33 | 0.00 | 0.00 | 0.00 |
| Phi-2-7B + CoT | 0.43 | 0.05 | 0.01 | 0.00 | 0.00 | - |
| Phi-2-7B + RoT | 0.54 | 0.16 | 0.01 | 0.01 | 0.00 | - |
| Phi-2-7B + CRE | **0.91** | **0.86** | **0.79** | 0.19 | 0.05 | 0.00 |
| Phi-2-7B + CreDes | - | - | - | **0.46** | **0.31** | **0.19** |
| Mistral-7B + RAP | 0.49 | 0.41 | 0.35 | 0.07 | 0.00 | 0.00 |
| Mistral-7B + CoT | 0.84 | 0.41 | 0.24 | 0.05 | 0.08 | - |
| Mistral-7B + RoT | 0.81 | 0.49 | 0.21 | 0.10 | 0.12 | - |
| Mistral-7B + CRE | **0.97** | **0.94** | **0.82** | 0.24 | 0.12 | 0.03 |
| Mistral-7B + CreDes | - | - | - | **0.54** | **0.37** | **0.21** |
| Mixtral-8x7B + RAP | 0.49 | 0.44 | 0.35 | 0.15 | 0.04 | 0.00 |
| Mixtral-8x7B + CoT | 0.81 | 0.63 | 0.55 | 0.18 | 0.20 | - |
| Mixtral-8x7B + RoT | 0.87 | 0.71 | 0.55 | 0.29 | 0.27 | - |
| Mixtral-8x7B + CRE | **0.99** | **0.97** | **0.93** | 0.34 | 0.22 | 0.13 |
| Mixtral-8x7B + CreDes | - | - | - | **0.75** | **0.57** | **0.40** |

Table 2: Accuracy under GSM8K

| Model | RAP | RoT | CoT | **CRE** |
|---|---|---|---|---|
| Llama-2-7B | 0.51 | 0.54 | 0.47 | **0.92** |
| Phi-2-7B | 0.45 | 0.48 | 0.48 | **0.89** |
| Mistral-7B | 0.39 | 0.32 | 0.31 | **0.85** |
| Mixtral-8x7B | 0.48 | 0.50 | 0.49 | **0.90** |

### 4.5 Discussion

This study introduced the CreDes framework, which combines CRE and DES to improve LLMs' ability to handle long-range reasoning tasks. CRE ensures robust causal relationships between reasoning steps, and DES can lower the complexity of long-range reasoning by using a bidirectional search approach. Our experiments, particularly in the Blocksworld and Hanoi Tower scenarios,

Table 3: Succcess Rate under Hanoi Tower

| Model | 3-step | 5-step | 7-step | 9-step | 11-step | 13-step |
|---|---|---|---|---|---|---|
| Llama-2-70B + RAP | 0.57 | 0.42 | 0.22 | 0.07 | - | - |
| Llama-2-7B + RAP | 0.29 | 0.21 | 0.11 | 0.00 | - | - |
| Llama-2-7B + CoT | 0.34 | 0.23 | 0.10 | 0.02 | 0.00 | 0.00 |
| Llama-2-7B + RoT | 0.41 | 0.27 | 0.13 | 0.04 | - | - |
| Llama-2-7B + CRE | **0.45** | **0.39** | **0.24** | 0.12 | 0.01 | 0.00 |
| Llama-2-7B + CreDes | - | - | - | **0.27** | **0.14** | **0.07** |
| Phi-2-7B + RAP | 0.27 | 0.21 | 0.14 | 0.01 | - | - |
| Phi-2-7B + CoT | 0.33 | 0.0.22 | 0.10 | 0.02 | 0.00 | 0.00 |
| Phi-2-7B + RoT | 0.24 | 0.12 | 0.02 | 0.00 | - | - |
| Phi-2-7B + CRE | **0.40** | **0.25** | **0.17** | 0.03 | 0.00 | 0.00 |
| Phi-2-7B + CreDes | - | - | - | **0.33** | **0.20** | **0.09** |
| Mistral-7B + RAP | 0.34 | 0.25 | 0.14 | 0.04 | - | - |
| Mistral-7B + CoT | 0.40 | 0.32 | 0.21 | 0.09 | 0.00 | 0.00 |
| Mistral-7B + RoT | 0.35 | 0.22 | 0.17 | 0.02 | - | - |
| Mistral-7B + CRE | **0.49** | **0.37** | **0.26** | 0.15 | 0.03 | 0.00 |
| Mistral-7B + CreDes | - | - | - | **0.37** | **0.19** | **0.11** |
| Mixtral-8x7B + RAP | 0.40 | 0.24 | 0.15 | 0.06 | - | - |
| Mixtral-8x7B + CoT | 0.45 | 0.27 | 0.14 | 0.02 | 0.00 | 0.00 |
| Mixtral-8x7B + RoT | 0.37 | 0.22 | 0.10 | 0.00 | - | - |
| Mixtral-8x7B + CRE | **0.50** | **0.35** | **0.22** | 0.11 | 0.01 | 0.00 |
| Mixtral-8x7B + CreDes | - | - | - | **0.42** | **0.25** | **0.12** |

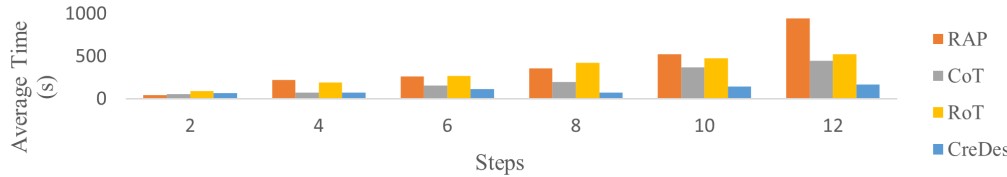

Figure 3: Improvement in reasoning speed for long-range tasks (based on a single A100 GPU).

demonstrated significant improvements in accuracy and efficiency over existing methods, implying that CreDes can effectively address the problem of causal hallucinations and huge search spaces.

### 4.6 Limitation

In scenarios with strict order of precedence, such as the Hanoi Tower, the accuracy is significantly lower compared to tasks like Blocksworld. The DES approach, while effective for moderate-length tasks, struggles with very long reasoning steps, leading to a decline in performance. Additionally, maintaining causal logic through CRE and DES introduces computational overhead, which may limit the framework's scalability and applicability in real-world scenarios with limited resources. Finally, our approach pays insufficient attention to the sequential ordering of steps, and the ATE can only determine whether the causal logic makes sense, rather than recognizing, for example, the assumption encountered in the Hanoi Tower problem that the larger disk must be placed under the smaller disk.

## 5 Conclusion

By integrating CRE and DES, the CreDes framework has significantly advanced LLMs' capabilities in long-range reasoning tasks. This combined approach enhances the accuracy and efficiency of multi-step reasoning and maintains the problem-solving and reasoning abilities of pre-trained models across different tasks. Future work will focus on refining the framework to improve scalability and efficiency in various complex problem-solving scenarios.

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

# A  Appendix

## A.1  Validation Results of Model's Inherent Capabilities

To verify the success rate of our CRE method on other baseline tasks, we designed a control experiment to ensure that our approach does not impair the model's inherent problem-solving and reasoning abilities. Since DES is specifically designed for Blocksworld, a task with longer reasoning steps, the control experiments listed do not involve such lengthy reasoning steps; therefore, DES's performance is not tested in this section. The experimental results indicate that the CRE method can, to some extent, enhance the model's problem-solving capabilities on other baseline tasks without causing any reduction in performance. See Table 4.

Table 4: Results of model's inherent capabilities

| Model | AQUA | QASC |
|---|---|---|
| Llama-2-7B | 0.25 | 0.17 |
| Llama-2-7B + CRE | **0.74** | **0.62** |
| Baichuan-7B | 0.31 | 0.07 |
| Baichuan-7B + CRE | **0.85** | **0.31** |
| Mpt-7B | 0.11 | 0.05 |
| Mpt-7B + CRE | **0.65** | **0.27** |
| TAIDE-LX-7B | 0.27 | 0.21 |
| TAIDE-LX-7B + CRE | **0.89** | **0.72** |
| Qwen1.5-7B | 0.57 | 0.09 |
| Qwen1.5-7B + CRE | **0.75** | **0.37** |

## A.2  A Note on the Hanoi Tower Dataset

We generated and produced the Hanoi Tower dataset in the paper. The production method is to randomly generate several states conforming to the placement rules of the Hanoi Tower based on a given number of rods and disks, e.g., three rods and three disks, and randomly select one of these states as the starting and target states for a single sample. For a single sample, the classical partition algorithm is used to derive the pathway, and according to the length of the pathway, the sample is categorized into different number of steps groups, e.g., 3-steps, 5-steps, 7-steps, and so on. An odd number is chosen for the allocation because the most complex solving step of Hanoi Tower in the case of three rods and $n$ disks is $2^n - 1$ steps. We generated the dataset Hanoi Tower using exactly the same storage format and Prompt structure as Blocksworld and GSM8K.

## A.3  Prompt Templates Used During Training and Testing of CRE

---
**Prompt 1** Prompt Templates Used During **Training**
---
1: **Input:** Initial State ‖ Goal State **####** Pathway
2: **Output: ####** Pathway
3: **Pathway:** <Step1><Step2><Step3><step4>

---

---
**Prompt 2** Prompt Templates Used During **Testing**
---
1: **Input:** Initial State ‖ Goal State
2: **Output: ####** Pathway
3: **Pathway:** <Step1><Step2><Step3><step4>

---

 **A.4 Full Experimental Results under The Blocksworld Dataset**

Table 5: Succcess Rate under Blocksworld (Cont'd Table)

| Model | 2-step | 4-step | 6-step | 8-step | 10-step | 12-step |
|---|---|---|---|---|---|---|
| Baichuan-7B + RAP | 0.61 | 0.72 | 0.70 | 0.43 | 0.09 | 0.01 |
| Baichuan-7B + CRE | **0.93** | **0.74** | **0.71** | 0.25 | 0.05 | 0.00 |
| Baichuan-7B + CreDes | - | - | - | **0.63** | **0.47** | **0.29** |
| Mpt-7B + RAP | 0.25 | 0.06 | 0.00 | 0.00 | 0.00 | 0.00 |
| Mpt-7B + CRE | 0.32 | 0.11 | 0.04 | 0.00 | 0.00 | 0.00 |
| Mpt-7B + CreDes | - | - | - | 0.05 | 0.00 | 0.00 |
| TAIDE-LX-7B + RAP | 0.62 | 0.67 | 0.65 | 0.52 | 0.07 | 0.00 |
| TAIDE-LX-7B + CRE | **0.99** | **0.89** | **0.81** | 0.34 | 0.04 | 0.00 |
| TAIDE-LX-7B + CreDes | - | - | - | **0.70** | **0.54** | **0.35** |
| Qwen1.5-7B + RAP | 0.57 | 0.64 | 0.61 | 0.28 | 0.02 | 0.00 |
| Qwen1.5-7B + CRE | **0.92** | **0.77** | **0.73** | 0.34 | 0.08 | 0.02 |
| Qwen1.5-7B + CreDes | - | - | - | **0.61** | **0.46** | **0.36** |

