# OpenReview forum: "CreDes: Causal Reasoning Enhancement and Dual-End Searching for Solving Long-Range Reasoning Problems using LLMs"
_NeurIPS.cc/2024/Conference — Submitted to NeurIPS 2024_

### Official Review · Reviewer_BKzM · 2024-07-09

**Soundness:** 2
**Presentation:** 2
**Contribution:** 2
**Rating:** 5
**Confidence:** 3

**Summary:**

This paper develops a structured and generalized reasoning framework, CreDes, for long-range reasoning in LLMs. In the framework, the Causal Relationship Enhancement (CRE) is used to guarantee the solid causal rightness between each step of reasoning and state transition, and the Dual-End Searching (DES) approach is proposed to seek solutions by simultaneously starting from both the initial and goal states on the causal probability tree, to improve the efficiency.

**Strengths:**

1. This paper is well-structured and clearly states the problem they studied. It considers the long-range reasoning of LLMs from two aspects: the correctness from one-step reasoning (OSR) to state transition, and the efficiency of the solving process.
2. This paper transits the long-range reasoning problem of LLMs into the construction of causal probability trees from the initial and goal states and uses Dual-End Searching to improve efficiency. This is a reasonable and interesting thought.
3. The experimental results are SOTA in long-range reasoning tasks in terms of both accuracy and time efficiency.

**Weaknesses:**

1. The main concern is the understanding of ATE. This paper frequently uses ATE as part of the loss function and thinks the lower ATE can guarantee the solid causal rightness between each step of reasoning and state transition. However, ATE is used to measure the causal influence level between variables from the observational data, and causality does not mean rightness.
2. The DES section is not clear enough. It is suggested that more explanation be provided for the reason for the ATE as part of the loss. For example, if “B is the number of unfolded layers where the current leaf is located Ni”, what does E(A|do(B)) and E(A) mean in Formula (5)?
3. This paper needs to supplement the usage scenarios of methods, specifically in which scenario to use CreDes, in which scenario to use Cre alone, and whether Des is used separately.

**Questions:**

CRE:
1. Row 136-142: Do you assume that only the transition from correct OSR to correct state is causal, and the other three incorrect scenarios are non-causal? This is not the case. Generally, the determination of causality is based on observational data rather than subjective perception.
2. Row 145: “enhancing the causality of state transitions” seems to contradict the subsequent mention of lower |ATE|.
3. Row 146-147: Does the phrase “suppression of hallucinations is handled by the |ATE| loss” refer to reducing the causal relationship in the three wrong scenarios of incorrect OSR to state transitions? If so, is the causality in the correct category also weakened in the process?
4. Row 165-167: What is the basis for lower |ATE|? Normally, a low |ATE| does not signify a robust causal relationship but rather indicates a weak influence from X to Y.
5. Row 177-178: How to achieve “correct answers with strong causal effects|, and wrong answers with weak causal effects”?

DES:
1. Row 198: If the ATE is part of the loss function to be minimized, how could A and B have a strong causal effect?
2. Row 199: Please explain the detailed process of counterfactual assessments.
3. Row 204-205: Cannot understand why minimize ATE(A). Is there a tradeoff between tree unfolding and distance reduction? Can you explain it in an intuitive way?
4. Algorithm 1 step 5: What is the relationship between the “unfolded layers” of causal probability tree and the steps in “for every four steps do”? How to determine four steps rather than two steps?
5. Algorithm 1 step 6: Please explain the detailed process of “Determine intermediate steps and fill in details”.

Experiment:
1. Row 235-236: What is the training set? Please explain the training details.
2. Row 253-263: Is the 2-12 steps the shortest step for one problem in Blocksworld? How to determine the steps should be taken for one problem in Blocksworld?

**Limitations:**

None.

---

> ### Author Rebuttal · Authors · 2024-08-06
>
> Dear Reviewer BKzM,
>
> We want to express our gratitude for the thorough review and the constructive feedback on our paper.
>
> # Weaknesses:
> 1.Causality does not guarantee correctness; ATE constraints ensure causal consistency between OSR and the next state, while cross-entropy control path choose correctness.
>
> 2.We explained this in lines 191-192. For your convenience, we explain further: we evaluate the causal relationship between OSR and state transitions with the help of a similar idea in the CRE section, where we assess the causal relationship between tree unfolding hierarchy and distance between object blocks between states, where distance refers to the sum of the distances between the positions of each block of the object in the current state and the positions of each block of the object in the target state The distances are computed using Euclidean distances. This is what is meant by calculating the ATE between A  and B. We clarify that the way ATE(A) is written may have caused misunderstandings, and we revise it to ATE(A, B).
>
> 3.DES is unnecessary within the advantage interval (e.g., below 6 steps). DES rounds up long problems into the advantage interval. CRE is an internal oversight mechanism for the model’s reasoning process, while DES compensates for the model's shortcomings in long process tasks. When task difficulty exceeds the model's inherent performance, DES ensures reasoning within the advantage interval.
>
> # Questions:
> ## CRE:
> 1.We share your view and do not believe that only the transition from the correct OSR to the proper state is causal, while the other three incorrect scenarios are non-causal. However, we would like to clarify that the example indicated by the absolute drop in Fig. 2 ATE is causal. We are completing data observations based on multiple independent repetitive experiments with the same question asked repeatedly to observe the model's results.
>
> 2.It's not a contradiction. The idea is that for multiple independent repetitions of the experiment, the causal hallucinations in the model output are in the minority. Taking the hallucination scenario listed in Fig. 2 as an example, when the distribution of experimental results keeps changing (i.e., when hallucinations are still produced erratically), the absolute value of ATE is increasing, and when the distribution of experimental results tends to be stable (i.e., when the results tend to be stable and unchanged under different inference paths), the absolute value of ATE is decreasing. That is, after the cross-entropy selection path, the constraint by the decreasing absolute value of ATE makes the model state transition output results tend to be stable.
>
> 3.Not the three error scenarios, we clarify that the illustration figure may cause a misunderstanding; we want to express to reduce the occurrence of all causal error scenarios under a single path choice, where the causal error scenarios depend on the choice of path. The correctness of the categories has been proven by previous work that cross-entropy is effective enough, so the synergistic optimization of both, with simultaneous reduction, has the effect of converging to causally correct scenarios under the proper categories.
>
> 4.You can refer to our previous explanation.
>
> 5.You can refer to our previous explanation.
>
> ## DES:
> 1.The cross-entropy determines the OSR, but the calculation of the distance needs to ensure that the next state is accurate, so the same logic as in the CRE section, what we need to ensure is that the entire distribution of observations is stable, i.e., that the phantom scenarios between the OSR and the output of the following state are eliminated as much as possible, and that the more complex the distribution is (and the more scenarios there are), the larger the ATE is.
>
> 2.For example, the unfolding direction is to move the red block from the left to the right, and the counterfactual is the observation when this is not done or when the move is to another color block. Since the training phase produces more hallucinatory output, these observations can help us somewhat with the counterfactual assessment.
>
> 3.We want the tree to unfold in a helpful direction for the solution, i.e., the head tree unfolds in the Goal State. In contrast, the tail tree unfolds in the Initial State, and we want the tree unfolding and the distance reduction to be positively correlated or even strongly causal. As shown in the second half of Fig. 1, we will compute the Euclidean distance between the current position and the goal position for each block's coordinates, which in turn makes all object blocks move towards the goal.
>
> 4.Each layer unfolded represents one step out of every 4 steps done. The model is a one-time output of 4-steps; only when determining the target state for the model to achieve in the first 4-steps will we first search; that is, the model generates several alternative 4-step states based on the Initial State and then selects the optimal solution from them. 4-steps can be done, and so can 2 or 3 steps, only that the 4-step efficiency will be higher. The time consumption is less under the condition of guaranteeing accuracy. If the modeling ability is further improved, excluding extended 6-step reasoning may be possible.
>
> 5.Determining states at steps 4 and 8, then reasoning from step 4 to 8. Experimental accuracy drop is due to the need to improve the success rate in this process.
>
> # Experiment
> 1.Training and test sets are mutually exclusive, randomly sampled from the original dataset. Default hyperparameters of the pre-trained model were used, with inputs and outputs detailed in the Appendix and discussed in the response to Reviewer yRix.
>
> 2.The shortest step, because there can be many ineffective repetitive steps, such as repeatedly moving the red block left and right. Determine the steps needed as given by the data set.
>
> Thank you for your time and consideration.

---

> ### Author Response · Authors · 2024-08-11
>
> Dear Reviewer BKzM,
>
> We have submitted our rebuttal several days ago, and the discussion process is now more than halfway through. However, we have not yet received your response. We greatly value your insights and are eager to hear your further feedback on our paper. Your comments will be instrumental in helping us improve the manuscript.
>
> Thank you for your time and consideration.
>
> Sincerely,
> Authors

---

> ### Comment · Reviewer_BKzM · 2024-08-12
>
> Thanks for the detailed responses. The submission's statement suggests that the goal is to reduce the causal influence of OSR on state transitions, thereby making the results stable and unchanged under different inference paths. However, several issues require further clarification:
>
> 1. The Concept of Causality. The concept of "causality" is repeatedly mentioned in the paper, like "ensure the causality between OSR and state transition in LLMs," including the causality measure between OSR and state transition," enhancing the causality of state transitions," etc. Combined with the feedback in the rebuttal, does it mean "distribution of experimental results tends to be stable" is equivalent to "enhancing the causality"? If these two concepts are equivalent, please provide relevant references to support this equivalence.
>
> 2. The relationship between "Enhancing causality" and "Lower |ATE|". ATE is an estimand for causal effect estimation. "Low |ATE|" typically indicates a non-significant causal effect, what is the relationship between "lower |ATE|" and "enhancing causality"? The statement in the rebuttal, "we would like to clarify that the example indicated by the absolute drop in Fig. 2 ATE is causal" also raises this issue. If there is additional clarification regarding this, please illustrate with equations and references.
>
> 3. The Benefits of Stability of Results. Setting aside the causal effect estimand ATE, in the rebuttal, does "the results tend to be stable and unchanged under different inference paths" mean that the correctness or incorrectness of OSR has a minimal causal impact on state transition outcomes? What are the advantages of this approach? Is the stability of results inherently beneficial? If so, please provide references to support this.

---

> > ### Author Response · Authors · 2024-08-12
> > **To Reviewer BkzM and AC (Part 1)**
> >
> > Dear AC and Reviewer BKzM,
> >
> > We sincerely appreciate the insightful feedback provided by the reviewer. It has helped us recognize potential misunderstandings that might exist among the reviewers.
> >
> > We kindly request that the Area Chair share this QA exchange with all the reviewers to ensure these clarifications are understood broadly. Additionally, the 5000-word limit for the rebuttal is significantly constraining, and we respectfully request permission to exceed this limit to provide a more comprehensive response.
> >
> > # Clarification on Our Perspective
> >
> > Firstly, we urge the reviewer to revisit our rebuttal to the first point raised under "Weaknesses." We apologize for any lack of clarity in our initial response due to the word limit.
> >
> > We must clarify that the reviewer's position is fundamentally different from the statement attributed to us: "The goal is to reduce the causal influence of OSR on state transitions, thereby making the results stable and unchanged under different inference paths." The reviewer's subsequent questions stem from this misunderstanding. We will now present a complete theoretical explanation, which we believe will address the three concerns that arose from this misinterpretation.
> >
> > # Theoretical Background and Practical Considerations
> >
> > In typical scenarios, the ATE (Average Treatment Effect) is expressed as follows:
> >
> > $ \text{ATE} = \mathbb{E}[Y(1) - Y(0)] $
> >
> > Isolating this formula in a purely metaphysical sense could lead to questions similar to the reviewer's. However, the context differs significantly when applied to large model inference. We perform numerous output trials to calibrate the model during the training process. From our experimental results, these output samples demonstrate a variety of possibilities, such as:
> >
> > Type 1: Certain samples are challenging to answer correctly, regardless of training, resulting in near-random correct/incorrect states.
> >
> > Type 2: There is a positive correlation between epoch count and correct answer frequency for some samples, significantly when aided by standard training techniques like RAP and CoT.
> >
> > Type 3: Some samples can be answered correctly with minimal training, showing no correlation between epoch count and correct answer frequency.
> >
> > As Blocksworld researchers, the current goal is to maximize the correct rate of Type 1 samples, effectively converting more Type 1 samples into Type 2.
> >
> > # Explanation of Our Approach
> > It is crucial to note that we do not train for a specific problem in each training cycle but for all samples in the training set. Therefore, we must distinguish between aggregate and individual treatment effects. Suppose we define an individual-level treatment effect evaluation $ \tau_i $;  it can be written as:
> >
> > $  \tau_i = Y_i(1) - Y_i(0)  $
> >
> > $
> > \text{ATE} = \frac{1}{N} \sum_{i=1}^{N} \tau_i
> >  $
> >
> > Among them:
> >
> > -$ N $ is the number of individuals in the sample.
> >
> > -$ \tau_i $ represents the processing effect of the $i $ th individual.
> >
> > Given that large language models possess basic logical reasoning abilities (as discussed in our rebuttal), our objective is to enhance rather than reconstruct this capability. Our experimental data supports this (we acknowledge that we did not include this in the appendix but will include relevant visualizations in the revised submission). For repeated experiments on a single sample, the model's responses follow a normal distribution.
> >
> > $
> > \tau_i \sim N(\mu, \sigma^2)
> > $
> >
> > Note that this involves a complete count of all possible answers, not just a binary correct/incorrect classification. The correct response occurs most frequently.
> >
> > In this context, the mean of the normal distribution aligns with the expression for individual-level ATE, while the variance reflects the variability of individual effects. Based on this, we propose the following logical extension:
> >
> > When individual effect consistency is high (low variance), the causal effect is more robust and consistent, resulting in a solid causal relationship even if the ATE is small.
> > When individual effect consistency is low (high variance), a large ATE does not necessarily indicate a robust causal relationship because the variability among individual responses may be significant.

---

> > ### Author Response · Authors · 2024-08-13
> >
> > Dear Reviewer BKzM,
> >
> > We would like to express our sincere gratitude for your previous responses and engagement with our paper. We noticed that you raised some further questions after our initial rebuttal, and we have since provided additional responses to address your concerns.
> >
> > We wanted to check in to see if our latest responses have resolved your queries, and to ask if there are any remaining questions or issues that you would like us to address. We apologize for reaching out so directly, but with the discussion period nearing its end, we will soon be unable to participate in further discussion.
> >
> > Once again, thank you very much for your time and thoughtful feedback. We greatly appreciate your support and look forward to hearing from you.
> >
> > Best regards,
> >
> > Authors

---

> > > ### Comment · Reviewer_BKzM · 2024-08-14
> > >
> > > Thanks for the detailed responses, which have solved my major concerns. I will raise my rating accordingly.
> > >
> > > Besides, understand the goal of achieving convergence in your results. In the submission, it may be necessary to differentiate and elucidate the concepts of ATE and variance for ITE. Additionally, it is important to distinguish between the significance of the ATE, as discussed in the provided references, and the value of the ATE itself. I suggest the authors continue to improve this submission.

---

> ### Author Response · Authors · 2024-08-12
> **To Reviewer BkzM and AC (Part 2)**
>
> # Detailed Explanation
> When $\sigma^2$  (variance) is small, the treatment effects for all individuals are very close to the mean $\mu $. This suggests that the treatment effect is highly consistent and stable across different individuals:
>
> Causal Certainty: Due to the high consistency of individual effects, the treatment’s impact can be considered specific rather than incidental or person-dependent. This consistency is usually regarded as a hallmark of a strong causal relationship.
>
> Robustness of Causal Effects: Even if the average effect (ATE) is small, this consistency indicates that the treatment has a similar impact on all individuals, making the causal relationship robust and strong.
>
> Conversely, when  $\sigma^2$ is large, the treatment effects among different individuals may vary significantly, leading to a high level of inconsistency:
>
> Causal Uncertainty: The treatment effect may manifest as a strong positive effect in some individuals and a negative effect in others. This high variability implies that the treatment's impact is uncertain, leading to a weaker causal relationship.
>
> Inconsistency of Causal Effects: Even with a large average effect (ATE), the significant differences in individual responses make it difficult to assert that the treatment effect is consistent across all individuals. Consequently, such a causal relationship is generally considered inconsistent or weak.
>
> $
> \sigma^2 = \frac{1}{N} \sum_{i=1}^{N} (\tau_i - \text{ATE})^2
>  $
>
> # Practical Example and Application to Our Work
> To better understand this extended perspective, let's consider a specific example:
>
> Take the Blocksworld scenario as an example. At the beginning of training with a fixed OSR, various state transition results such as A, B, C, D, and E might indicate low consistency. By using methods like fine-tuning that rely on the model's cross-entropy loss, the model can be constrained to produce fewer state transition results, such as only A, B, and C. However, more is needed, as the ultimate goal of using ATE is to constrain the state transition results to only one outcome, specifically A.
>
> Towards the end of a training process, the following two situations may occur:
> Assume we plot A, B, C, D, and E on the X-axis and the frequency of the corresponding output samples on the Y-axis and observe the distribution. Our objective is to make the distribution tend toward one with low variability.
>
> Scenario A: High Consistency, Low ATE: In this case, the variability of the distribution is small, meaning fewer types of situations occur.
>
> Scenario B: Low Consistency, High ATE: In this case, the variability of the distribution is large, meaning more types of situations occur.
>
> From here, we begin to consider the issue of consistency. These two scenarios have commonalities with the model training process. The combination of OSR and state transitions is not unique for a single sample under 2-class- and 3-class samples. Therefore, we aim to achieve a one-to-one correspondence between OSR and state transitions. Note that a one-to-one correspondence does not necessarily mean the combination of OSR and state transitions is correct. However, during the training process of a large model, the model's cross-entropy loss function also plays a role. Just as with methods like RAP and CoT that freeze the model, the model's cross-entropy loss function alone can somewhat control consistency. Additionally, because our model is not frozen, the output consistency of the model improves with each epoch during training. Therefore, our training process is an optimization process from Scenario B to Scenario A. ATE does not work in isolation but cooperates with the cross-entropy loss function.
>
> The dimensions of cross-entropy loss and ATE are different. In the early stages of model training, the changes in cross-entropy loss (or PPL) are dramatic, far exceeding the changes in ATE. As a result, ATE plays a minor role at this stage. Only in the later stages of training when the cross-entropy loss decreases to a certain extent, and its rate of change approaches that of ATE, ATE begins to play a role, assisting the model in further optimization.
>
> The above discussion focuses on a single-sample perspective. However, to evaluate the overall model training process across epochs, we assess the model by its accuracy in answering validation set questions, not by ATE.
> # Conclusion
> We believe that this reviewer's comments convey the same concerns that other reviewers may have, so we hope everyone can take a look.
>
> In summary, the method presented in our paper involves fine-tuning during the training process, and the reviewer's perspective might have been overly broad.
>
> Meanwhile, we have attached a relevant reference for your further understanding:
>
> [1] Bao G, Zhang H, Yang L, et al. Llms with chain-of-thought are non-causal reasoners[J]. arXiv preprint arXiv:2402.16048, 2024.
>
> Please review the 4 and 5 sections.
>
> Thank you once again for your efforts and consideration.

---

> ### Author Response · Authors · 2024-08-14
>
> Dear Reviewer BKzM,
>
> Thank you very much for your thoughtful feedback and for taking the time to carefully review our responses. We are pleased to hear that our explanations have resolved your major concerns, and we sincerely appreciate your willingness to raise your rating accordingly.
>
> We also value your insightful suggestions regarding the differentiation and elucidation of the concepts of ATE and variance for ITE, as well as the distinction between the significance and the value of ATE. We will certainly incorporate these points into our revised submission to further improve the clarity and impact of our work.
>
> Once again, thank you for your constructive feedback and support. Your comments have been extremely helpful, and we are committed to making the necessary revisions to enhance our paper.
>
> Best regards,
>
> Authors

---

### Official Review · Reviewer_yRix · 2024-07-10

**Soundness:** 3
**Presentation:** 2
**Contribution:** 3
**Rating:** 5
**Confidence:** 3

**Summary:**

This paper introduces CreDes, a framework to improve the long-range reasoning capabilities of LLMs, consisting of two main components: Causal Relationship Enhancement (CRE) and Dual-End Searching (DES). CRE is developed to reduce causal hallucinations in LLMs by strengthening the causal relationships between reasoning steps and state transitions; it uses structural causal modeling and optimizes the Average Treatment Effect (ATE) during training. DES breaks down long-range reasoning tasks into shorter segments by simultaneously searching from both the initial and goal states on a causal probability tree. The authors evaluate CreDes on Blocksworld, GSM8K, and Hanoi Tower puzzles, showing improvements in both accuracy and efficiency compared to existing methods.

**Strengths:**

- CreDes demonstrates significant improvements over existing methods, especially for complex tasks requiring many reasoning steps.
- The use of causal modeling concepts like ATE provides a solid theoretical foundation for the proposed approach.
- The method shows effectiveness across different types of reasoning tasks (e.g., spatial reasoning, math problems).
- CreDes enables simultaneous multi-step reasoning, potentially reducing computation time compared to sequential methods.

**Weaknesses:**

Major concerns:

- The generalizability and scalability need better justification. The paper primarily tests the CreDes framework on Blocksworld, Hanoi Tower, and some mathematical reasoning tasks (GSM8K). These are relatively structured, rule-based problems that may not represent the full spectrum of reasoning challenges. In addition, the proposed method cannot be well scaled to long-range reasoning; for example, in Table 1, performance drops significantly for Blocksworld tasks beyond 8 steps, with success rates falling from 0.68 to 0.34 for 12-step problems using Llama-2-7B + CreDes. Table 3 shows even steeper declines for Hanoi Tower, with success rates dropping from 0.27 at 9 steps to just 0.07 at 13 steps for Llama-2-7B + CreDes. Notably, the authors explicitly acknowledge this limitation in Section 4.6, stating: "The DES approach, while effective for moderate-length tasks, struggles with very long reasoning steps, leading to a decline in performance."

- The presentation of this paper could be improved.

  -- In the problem definition, there is no explanation of the difference between training without common instructions and with common instructions.

  -- There is no detailed discussion of the differences between correlation and causation in Sec 3.2. I am confused about whether the correlation of two variables has anything to do with their distributions.

  -- While efficiency gains are mentioned, the added complexity of CRE and DES likely introduces some computational overhead, which could be further discussed.

  -- There is no analysis of the impact of the choices of hyperparameters on the methods, particularly in the CRE component.

- The proposed method lacks comparison to more recent state-of-the-art methods. The paper compares CreDes mainly to older baselines: Reasoning via Planning (RAP), Chain of Thought (CoT), and Reflexion of Thoughts (RoT). However, it doesn't evaluate against more recent advances in LLM reasoning, such as Tree of Thoughts (ToT) extensions in line 42, and the paper doesn't mention or compare to other recent works such as [a] and [b], which also address multi-step reasoning challenges. As a result, the technical contribution is not entirely clear.

[a] Chengrun Yang, Xuezhi Wang, Yifeng Lu, Hanxiao Liu, Quoc V Le, Denny Zhou, & Xinyun Chen (2024). Large Language Models as Optimizers. In The Twelfth International Conference on Learning Representations.

[b] Shibo Hao, Yi Gu, Haodi Ma, Joshua Jiahua Hong, Zhen Wang, Daisy Zhe Wang, & Zhiting Hu (2023). Reasoning with Language Model is Planning with World Model. In The 2023 Conference on Empirical Methods in Natural Language Processing.

Minor concerns:

- Experiments are primarily conducted with 7B parameter models, leaving questions about scalability to larger models. How does the performance of CreDes scale with increasing model size (e.g., to 10B+ parameters)? The computational overhead may limit the framework’s scalability and applicability in real-world scenarios with limited resources.

- The approach achieves significantly lower accuracy in tasks with very strict ordering constraints, such as the Hanoi Tower problem.

- Since Blocksworld involves random steps, an analysis of the robustness of the performance may be needed.

 - More analysis/discussion on the sequential ordering of steps may be helpful. Notably, the ATE cannot recognize casual logic.

 - Some editorial issues, e.g., Line 110

**Questions:**

Please see the weakness part.

**Limitations:**

The authors have discussed the limitations, and it is adequate to me. I do not see any potential negative societal impact.

---

> ### Author Rebuttal · Authors · 2024-08-06
>
> Dear Reviewer yRix,
>
> We want to express our gratitude for the thorough review and the constructive feedback on our paper.
>
> # Weaknesses:
> ## Major concerns:
> 1.(1)In terms of scalability, due to the faster reasoning, it is expected for tasks with high FPS rate requirements, such as unstructured autopilot tasks (turn left, turn right, etc.), and for actions with contextual causal logic reasoning relationships, such as gathering wood, digging up rocks, lighting torches, etc., as in the Minecraft example, it is expected in our current preliminary experiments to be realized. We initially plan to leverage the open-source Minecraft simulator from our previous work as a calibration tool for the long task decomposition steps we have planned, to assess the accuracy of our decomposition steps, and then to try to generalize our approach to open-world tasks, moving closer to the goal of embodied intelligence. We believe that simulation-based game simulations such as Minecraft have an expected better migration adaptation to the real world, but of course, there is more to it than that, and we also have a vision of developing simulators based on platforms such as Unreal to visualize decision-making process simulations.
>
> 1.(2)We recognize that this is the main limitation of our current work, partly because the middle gap portion formed after the DES bipartite search is insufficient for similar examples in the training dataset. The decrease in effectiveness is mainly due to the failure of the double-ended search (i.e., the inability to approach from both ends to the middle or the failure of inference in the middle gap part). We believe that further expansion of the dataset, such as additional generation of more training data that conforms to the rules, may be helpful for the overall performance improvement.
>
> 2.(1) Yes. Instruction Format: Our format differs from CoT's, designed per the pre-trained model publisher's manual and to differentiate input data parts accurately.
>
> 2.(2)Correlation and causality are different concepts; put, causality further enhances correlation. For example, if the sample regression equation is y = x, y and x are positively correlated. Still, it isn't very objective to say that there is strong causality between y and x because we cannot confirm whether a counterfactual sample that is not observed would still be able to be fitted to this equation.
>
> 2.(3) CRE increases training time compared to frozen models like RAP and CoT. However, RAP's complex Monte Carlo search tree and node inferences require more time. Our 7B model's inference speed is faster than 70B's. DES search expands smaller than Monte Carlo trees, as shown in our time statistics.
>
> 2.(4) We need to clarify that we did not perform hyperparameter optimization. The hyperparameters we used are the default options officially given by the open-source pre-trained models on the Huggingface website.
>
> 2.(5)  The first paper on prompt optimization uses the GSM8K dataset but differs in research focus. The second paper on RAP is a benchmark for comparison, referenced in our work.
>
> ## Minor concerns:
> 1.We acknowledge that our experiments were primarily conducted on 7B parameter models. The choice was made due to computational constraints and availability. The Mixtral-8x7B model is also larger than the 7B model so that it can be used as a reference for the performance of larger sizes. We conducted some additional experiments at the model size of 13B, but due to time constraints, only the following experimental results were obtained.
>
> ### Blocksworld
>  Model               | 2-step | 4-step | 6-step | 8-step | 10-step | 12-step
> -|-|-|-|-|-|-
>  Llama-2-13B+RAP     | 0.44   | 0.42   | 0.38   | 0.11   | 0.00    | 0.00
>  Llama-2-13B+CoT     | 0.51   | 0.63   | 0.39   | 0.29   | 0.07    | 0.00
>  Llama-2-13B+RoT     | 0.49   | 0.70   | 0.30   | 0.07   | 0.00    | 0.00
>  Llama-2-13B+CRE     | 0.95   | 0.82   | 0.74   | 0.25   | 0.07    | 0.00
>  Llama-2-13B+CreDes  | -      | -      | -      | 0.65   | 0.49    | 0.37
>
> ### GSM8K
>  Model      | RAP  | RoT  | CoT  | CRE
> -|-|-|-|-
>  Llama-2-13B| 0.50 | 0.57 | 0.49 | 0.93
>
> ### Hanoi Tower
>  Model              | 3-step | 5-step | 7-step | 9-step | 11-step | 13-step
> -|-|-|-|-|-|-
>  Llama-2-13B+RAP    | 0.30   | 0.20   | 0.12   | 0.00   | -       | -
>  Llama-2-13B+CoT    | 0.33   | 0.24   | 0.09   | 0.03   | 0.00    | 0.00
>  Llama-2-13B+RoT    | 0.44   | 0.30   | 0.12   | 0.03   | -       | -
>  Llama-2-13B+CRE    | 0.42   | 0.38   | 0.27   | 0.10   | 0.01    | 0.00
>  Llama-2-13B+CreDes | -      | -      | -      | 0.34   | 0.15    | 0.07
>
> From the results, there is not much difference between the experimental results under 13B and 7B, and we believe that the difference can be regarded as a random error generated by different training. From the performance comparison between the 70B model and the 7B model under the RAP method, the performance of the 70B model will be relatively improved. However, considering inference speed, the 70B model is much slower than the 7B, and it needs to be loaded with a certain amount of quantization, and the performance loss is equally present.
>
> 2.Hanoi Tower tasks are harder than Blocksworld's due to stricter constraints. The cause may be the 7B model's limited inference capacity and room for method optimization to control strict order.
>
> 3.Blocksworld generally does not involve random steps..
>
> 4.In Blocksworld, only top squares can move; middle or bottom squares cannot. Similarly, in the Tower of Hanoi, size discrimination prohibits larger blocks on smaller ones. Such states are considered solving failures.
>
> 5.We will adjust these issues in the revised version.
>
> We hope these clarifications address your concerns and improve the overall understanding of our work. We are committed to enhancing the paper based on your valuable feedback. Thank you for your time and consideration.

---

> ### Author Response · Authors · 2024-08-11
>
> Dear Reviewer yRix,
>
> We have submitted our rebuttal several days ago, and the discussion process is now more than halfway through. However, we have not yet received your response. We greatly value your insights and are eager to hear your further feedback on our paper. Your comments will be instrumental in helping us improve the manuscript.
>
> Thank you for your time and consideration.
>
> Sincerely,
> Authors

---

> > ### Comment · Reviewer_yRix · 2024-08-11
> >
> > I appreciate the author's detailed response.
> >
> > I would like to keep my score and believe that this paper is lightly above the acceptance bar.

---

> ### Author Response · Authors · 2024-08-12
>
> Dear Reviewer yRix,
>
> Thank you for taking the time to review our paper and for your thoughtful comments throughout the process. We greatly appreciate your recognition of our detailed responses and your continued support of our work.
>
> We respect your decision to maintain your score and are grateful that you consider our paper to be above the acceptance bar. Your feedback has been invaluable in refining our research, and we are hopeful that the contributions we have made will positively impact the field.
>
> Thank you once again for your efforts and consideration.
>
> Best regards,
>
> Authors

---

### Official Review · Reviewer_MmUh · 2024-07-11

**Soundness:** 2
**Presentation:** 2
**Contribution:** 2
**Rating:** 3
**Confidence:** 3

**Summary:**

The integration of Causal Relationship Enhancement (CRE) and Dual-End Searching (DES) mechanisms presents a novel solution to addressing causal hallucinations and large search spaces in long-range reasoning tasks. The CRE mechanism’s use of Structural Causal Modeling (SCM) and Average Treatment Effect (ATE) is  ensure causality between reasoning steps. Extensive testing on datasets such as Blocksworld, GSM8K, and Hanoi Tower demonstrates the effectiveness of the CreDes framework.

**Strengths:**

The idea seems novel and it test on well-known reasoning datasets.

**Weaknesses:**

The method presented in this paper evaluates ATE on LLMs, but this approach's validity hinges on the assumption that LLMs can perfectly represent the real-world environment. The very reason we criticize LLMs for their reasoning issues is because their inferences are not accurate. Estimating ATE might only bring the prediction results closer to Y while maximizing the influence of the intervention factor on Y. However, it does not necessarily mean that the intervention factor is the true cause of Y. In other words, since there is no alignment with the causal relationships in real-world scenarios, the implementation of this method does not prove that the reasoning is causally sound.

The method lacks deeper thinking. The authors just apply the concept of ATE to the Chain-of-Thought (CoT) without thorough analysis. This oversight leads to a misalignment between the experimental results and the motivation of the paper. Suppose LLMs are not a good s simulations of the real world. In that case,  performing interventions on LLMs (whether they align with the real world or their identifiability) requires sound theoretical analysis and experimental validation. The current paper lacks a deep discussion on this matter.

**Questions:**

1. How does DES ensure that it does not fall into local optima during the search process?
2. What are the computational requirements and limitations of using CreDes in practical applications?

**Limitations:**

See weakness.

---

> ### Author Rebuttal · Authors · 2024-08-06
>
> Dear Reviewer MmUh,
>
> We want to express our gratitude for the thorough review and the constructive feedback on our paper.
>
> # Weaknesses:
>
> ATE (Average Treatment Effect) is a metric used in causal inference to measure the average impact of a treatment or intervention on an outcome across a population. Specifically, ATE represents the difference in the average outcome between individuals who receive the treatment and those who do not.
>
> ATE is a reasonable metric capable of reflecting causality in causal inference. While real-world causal relationships are complex, and no single metric can capture all aspects perfectly, ATE is widely recognized as one of this domain's most reasonable and practical metrics. Although every metric has limitations, ATE has been validated and accepted by the research community for its robustness in estimating causal effects. We acknowledge the complexity of real-world causality and will consider incorporating other metrics in future work to enhance our causal analysis further.
>
> The current work focuses on highly structured tasks to demonstrate the effectiveness of the CreDes framework. We agree that exploring its applicability to less structured, dynamic tasks is essential.
>
> We are already working on designing related experiments based on open-world scenarios such as Minecraft and expect to argue further in a subsequent paper. We are currently actively designing experiments concerning the following related work: https://github.com/CraftJarvis
>
> [1]. Wang Z, Cai S, Liu A, et al. Jarvis-1: Open-world multi-task agents with memory-augmented multimodal language models[J]. arXiv preprint arXiv:2311.05997, 2023.
>
> [2]. Wang Z, Cai S, Chen G, et al. Describe, explain, plan and select: Interactive planning with large language models enables open-world multi-task agents[J]. arXiv preprint arXiv:2302.01560, 2023.
>
> We initially plan to leverage the open-source Minecraft simulator from our previous work as a calibration tool for the long task decomposition steps we have planned, to assess the accuracy of our decomposition steps, and then to try to generalize our approach to open-world tasks, moving closer to the goal of embodied intelligence.
>
> We believe that simulation-based game simulations such as Minecraft have an expected better migration adaptation to the real world, but of course, there is more to it than that, and we also have a vision of developing simulators based on platforms such as Unreal to visualize decision-making process simulations.
>
> The purpose of this paper is to eliminate the reasoning illusion of LLM. At the same time, ATE is a measure of causal significance, and the causality of LLM's reasoning at each step can be enhanced by combining ATE and LLM. Although this paper does not change LLM on the infrastructure, based on ATE and bidirectional search, it does improve the reasoning accuracy of LLM on some representative tasks. Thus, our experiments can support the motivation. Of course, we will consider improving LLM's infrastructure to enhance its fundamental reasoning ability. Due to our arithmetic limitations, we are currently tepid about conducting experiments such as LLM architecture tuning based on something other than open-source pre-trained models, but we will remain active.
>
> # Questions:
> 1.There is no guarantee, and our experiments confirm that DES searches are more efficient and have improved accuracy. The accuracy improvement is because we cut off long problems that are difficult for the model to solve into short issues that the model excels at, adjusting the problem to the advantageous range of the model's capabilities while ensuring that the search is relatively reliable with the appropriate search method.
>
> 2.This depends on the complexity of the actual application; for example, in the Minecraft example, for collecting wood, digging stones, lighting torches, and other actions that have contextual causal logic reasoning relationships, it is expected to be realized in our current preliminary experiments. We initially plan to leverage the open-source Minecraft simulator from our previous work as a calibration tool for the long task decomposition steps we have planned, to assess the accuracy of our decomposition steps, and then to try to generalize our approach to open-world tasks, moving closer to the goal of embodied intelligence. We believe that simulation-based game simulations such as Minecraft have an expected better migration adaptation to the real world, but of course, there is more to it than that, and we also have a vision of developing simulators based on platforms such as Unreal to visualize decision-making process simulations.
>
> We hope these clarifications address your concerns and improve the overall understanding of our work. We are committed to enhancing the paper based on your valuable feedback. Thank you for your time and consideration.

---

> > ### Comment · Area_Chair_jBaa · 2024-08-12
> >
> > Dear MmUh,
> >
> > Please acknowledge that you’ve reviewed the authors’ rebuttal and share any ongoing concerns. Do you feel that the authors’ explanations are still insufficient?
> >
> > Best,
> > AC

---

> ### Author Response · Authors · 2024-08-11
>
> Dear Reviewer MmUh,
>
> We have submitted our rebuttal several days ago, and the discussion process is now more than halfway through. However, we have not yet received your response. We greatly value your insights and are eager to hear your further feedback on our paper. Your comments will be instrumental in helping us improve the manuscript.
>
> Thank you for your time and consideration.
>
> Sincerely,
>
> Authors

---

### Official Review · Reviewer_Y4qM · 2024-07-12

**Soundness:** 2
**Presentation:** 2
**Contribution:** 2
**Rating:** 4
**Confidence:** 4

**Summary:**

This paper aims to improve LLMs in dealing with long-reason reasoning problems, especially the challenges of causal hallucination (inconsistency between one-step reasoning and corresponding state transition) and large search space. To tackle the first challenge, average causal effect of the one-step reasoning (treatment) on the state transition (outcome) is added to the loss function of the LLM; and for the second challenge, a dual-end (i.e. bi-directional) search approach is taken to improve efficiency. Experiments are conducted to demonstrate the effectiveness of the proposed method and its superiority over the compared existing methods.

**Strengths:**

1. An interesting idea of formalizing the problem from the perspective of causal effect and incorporating causal effect into the loss function.
2. The adoption of a dual-end search approach for improving efficiency.
3. The motivation of the paper is well presented in general.

**Weaknesses:**

1. The soundness of the proposed CRE method (for dealing with the challenge of causal hallucination) is in doubt.

(a) It's not clear why the method aims to $\textbf{minimize} the absolute value of the average treatment effect (ATE) of the one-step reasoning on state transition. Assuming that the ATE can be accurately estimated, what we want here would be to maximize the ATE that can be achieved by the LLM, i.e. when the one-step reasoning is correct done will likely lead to a correct state transition.

(b) It's not clear how an unbiased estimation of the ATE can be obtained, and what assumptions are made in terms of ATE estimation.

(c) The definition or understanding of ATE is incorrect. In particular, formula (2) is wrong, and formula (5) is incorrect too.

2. The presentation/technical quality requires improvement, including the presentation of related work. Please find below some examples:

(a) In Lines 42 to 44, it is said that the existing methods such as CoT are limited in task decomposition, but Lines 78-80 state that they can breakdown queries into manageable steps.

(b) Section 3.1 is titled as "Problem Definition", but it rather looks like a section on experiment setting.

(c) Lines 145-146 state that Fig. 1 shows "we leave the reasoning path selection to be controlled by the cross-entropy loss", but I cannot see this indicated in Fig. 2.

(d) Line 159: do(.) is an operator, specifically the do operator, rather do-calculus, although do-calculus uses this operator.

(e) Lines 159-160: the statement on the do(.) operator or do-calculus is incorrect, since an do operation on the treatment X would lead to the change of the outcome Y, especially if X is a cause of Y.

**Questions:**

1. How the ATE is estimated? Any assumptions, e.g. unconfoundedness, are made for the ATE estimation?
2. Line 166: what does "robust" mean here?
3. Why the model aims to minimize the absolute value of the ATE?

**Limitations:**

The authors have presented some discussions on the limitations of the proposed method. It would be better if the assumptions made could be presented more clearly and what the practical implications would be if the assumptions are violated.

---

> ### Author Rebuttal · Authors · 2024-08-06
>
> Dear Reviewer Y4qM,
>
> We want to express our gratitude for the thorough review and the constructive feedback on our paper.
>
> # Weaknesses:
>
> 1.(a):
>
> The concept is that, for multiple independent repetitions of the experiment, causal hallucinations in the model output are infrequent. Using the hallucination scenario depicted in Fig. 2 as an example, when the distribution of experimental results continues to fluctuate (i.e. when hallucinations are still produced sporadically), the absolute value of ATE increases. Conversely, when the distribution of experimental results stabilizes (i.e. when results become consistent and unchanging across different inference paths), the absolute value of ATE decreases. This means that after the cross-entropy selection path, the decreasing absolute value of ATE constrains the model's unexpected state transition, leading to more stable output results.
>
> 1.(b):
>
> Assumption 1: The causal illusions in the model output are few, i.e., the cases where causality between OSR and the next state is lost are few, as demonstrated in our experiments with practice when the pre-trained model can do so.
>
> Assumption 2: Equation (2) uses McNemar to test the significance of ATE to enhance the causal link between OSR and the next state, i.e., ATE decreases when OSR is the only cause of the next state.
>
> Assumption 3: The dataset given OSRs, i.e., the OSR of the desired reach and the OSR of the alternative paths, are mutually biased; for example, i.e., the OSR of the alternative paths is used as an intervention for the OSR of the desired reach path during the ATE evaluation process, and the OSR of the alternative paths is used as an intervention for the OSR of the alternative paths during the ATE evaluation process, of course, the paths' OSRs can be of many entries, for example The RAP method unfolded with Monte Carlo search will produce many alternative paths (the model is frozen), and the output of an untrained model is equally diverse.
>
> Assumption 4: During the ATE evaluation process, no attention is paid to the correctness of the OSR path selection, only to the strong causal between the OSR and the next state.
>
> 1.(c):
>
> Here are some references:
>
> [1] ANGRIST J, IMBENS G. Identification and Estimation of Local Average Treatment Effects[R/OL]//Econometrica,Econometrica. (2016-03). http://dx.doi.org/10.3386/t0118. DOI:10.3386/t0118.
>
> [2] Donald B Rubin. 1974. Estimating causal effects of treatments in randomized and nonrandomized studies.Journal of educational Psychology, 66(5):688.
>
> [3] Eric Nichols, Leo Gao, and Randy Gomez. 2020. Collaborative storytelling with large-scale neural language models. In Proceedings of the 13th ACM SIGGRAPH Conference on Motion, Interaction and Games, pages 1–10.
>
> According to the previous reference, Equation (2) is correct in using McNemar's test for the significance of the ATE.
> Equation (5) is designed to evaluate the causal link between the distance reduction and the number of tree unfolding layers, with the aim of better compressing the search space, and we clarify that this writing of ATE (A) may have caused a misunderstanding, and we will subsequently revise it to ATE (A, B)
>
> 2.(a) We revise paragraphs 78-80 to read: while COT makes sequence inference possible, its main contribution is interpretability, and the inference power is relatively limited. We will replace it in the revision.
>
> 2.(b) We will address this in our revision.
>
> 2.(c) We will address the issue of graph plotting in our revision, clarifying that the choice of paths can be controlled by cross-entropy, proven in the baseline method RAP. Our improvement is also not in proposing that cross-entropy controls the selection of paths but instead that it reduces the illusions in the paths after the paths have been chosen.
>
> 2.(d) All our do(.) refer to do-calculus.
>
> 2.(e)  We clarify that due to our writing error, you misunderstood that the do(.) operation on X results in a change in Y and that X is the cause of Y. In this paper, the interventions we impose on OSRs are, in fact, cross-references between desired and alternative OSRs, e.g., in computing the ATE of a desired OSR, several alternative OSRs will be interventions, as we state in the weakness. Thus, we minimize the ATE by stabilizing its distribution, not by simply enhancing the causal link between a particular OSR and the next state, but by enhancing the causal link between all OSRs and the next state, which is a holistic assessment process.
>
> # Questions:
> 1. As a further explanation, we refer you to the statement of weakness 1(a)(b); in our experimental data, when the model reaches a late stage of training, and the state transitions tend to stabilize (i.e., the number of state samples generating hallucinations decreases), the causal relationship between the OSR and the next state is strengthened, i.e., the OSR and the next state tend to be in one-to-one correspondence, which is different from what will happen in the RAP.
> 2. In this context, "robust" refers to the stability and reliability of the causal relationship between OSR and state transition under various conditions. You can refer to our related statements in the Weaknesses section.
> 3. The answer to this question is mentioned in our elaboration on weaknesses, which you can refer to in the weaknesses section.
>
> We hope these clarifications address your concerns and improve the overall understanding of our work. We are committed to enhancing the paper based on your valuable feedback. Thank you for your time and consideration.

---

> ### Author Response · Authors · 2024-08-11
>
> Dear Reviewer Y4qM,
>
> We have submitted our rebuttal several days ago, and the discussion process is now more than halfway through. However, we have not yet received your response. We greatly value your insights and are eager to hear your further feedback on our paper. Your comments will be instrumental in helping us improve the manuscript.
>
> Thank you for your time and consideration.
>
> Sincerely,
>
> Authors

---

> > ### Comment · Area_Chair_jBaa · 2024-08-12
> >
> > Dear Y4qM,
> >
> > Please acknowledge that you have read the authors’ rebuttal and express remaining concerns. Are the authors’ explanation insufficient? if so in what aspect? Please also ensure score rating to reflect your latest opinion of the paper.

---

> > ### Comment · Reviewer_Y4qM · 2024-08-13
> > **Thanks for your detailed responses**
> >
> > Thanks the authors for your detailed and helpful responses. I have more confidence on the soundness of the paper after reading your responses, and I have raised my score to 4.

---

> ### Author Response · Authors · 2024-08-13
>
> Dear Reviewer Y4qM,
>
> Thank you very much for your thorough review and for taking the time to carefully consider our responses. We are pleased to hear that our clarifications have increased your confidence in the soundness of our paper.
>
> We sincerely appreciate your willingness to re-evaluate our work and for raising your score. Your feedback has been instrumental in helping us improve our research, and we value the opportunity to learn from your insights.
>
> Thank you once again for your thoughtful review and consideration.
>
> Best regards,
>
> Authors

---

### Official Review · Reviewer_nWAC · 2024-07-13

**Soundness:** 2
**Presentation:** 3
**Contribution:** 2
**Rating:** 5
**Confidence:** 4

**Summary:**

This paper introduces a new framework, CreDes, designed to enhance causal reasoning in large language models (LLMs) and solve complex, long-range reasoning problems. The framework integrates two main innovations: the Causal Relationship Enhancement (CRE) mechanism, which applies cause-effect interventions to maintain causal accuracy across reasoning steps, and the Dual-End Searching (DES) method, which approaches problem-solving by initiating searches from both the initial and goal states to efficiently navigate large search spaces. The efficacy of CreDes is demonstrated through rigorous testing on challenging datasets like Blocksworld and Hanoi Tower, where it outperforms existing state-of-the-art models in both accuracy and efficiency.

**Strengths:**

1. Novel approach: The paper addresses essential limitations in LLMs' reasoning capabilities for long-range tasks in a causal perspective.
2. Comprehensive evaluation: The authors test their method on multiple datasets and compare against several baselines and shows improvements in both accuracy and time efficiency.

**Weaknesses:**

1. Limited model sizes: The experiments are primarily conducted on 7B parameter models, which may not reflect performance on larger state-of-the-art LLMs.
2. Lack of error analysis: The paper doesn't provide a detailed analysis of the types of errors made by the model or how they differ from baseline methods.
3. Dataset validity and construction: More details is needed for the use of a custom-made Hanoi Tower dataset which potentially limiting the reproducibility and generalizability of the results.
4. Computational efficiency and scalability: As mentioned in the Limitation, the paper lacks a detailed discussion of the computational requirements and scalability of the CreDes framework.
5. Generalization to less structured tasks: The framework's effectiveness is primarily demonstrated on highly structured tasks but it's unclear about its applicability to more dynamic or open-ended reasoning scenarios.
6. Lack of statistical significance: The paper doesn't report error bars or statistical significance for its experimental results.

**Questions:**

1. Given that experiments were conducted on 7B parameter models (Section 4.2), how does the performance of CreDes scale with larger model sizes? Have you tested it on models larger than 7B parameters.
2. Regarding the Hanoi Tower dataset described in Appendix, can you provide more details on how it was validated? How does it compare to established benchmarks in testing causal reasoning capabilities?
3. Figure 3 shows an unusual lack of variation in reasoning speed as the number of steps increases for the CreDes (blue bar). Can you explain this phenomenon and discuss its implications?
4. The paper focuses on structured tasks like Blocksworld and Hanoi Tower (Section 4.1). How might CreDes be adapted or applied to less structured reasoning tasks that don't have clear initial and goal states?
5. It's mentioned that CreDes can perform simultaneous multi-step reasoning. Can you elaborate on how this works and provide examples?

**Limitations:**

Yes

---

> ### Author Rebuttal · Authors · 2024-08-06
>
> Dear Reviewer nWAC,
>
> We want to express our gratitude for the thorough review and the constructive feedback on our paper.
>
> # Weaknesses:
> ## 1.Limited Model Sizes:
> We acknowledge that our experiments were primarily conducted on 7B parameter models. The choice was made due to computational constraints and availability. The Mixtral-8x7B model is also larger than the 7B model so that it can be used as a reference for the performance of larger sizes. We conducted some additional experiments at the model size of 13B, but due to time constraints, only the following experimental results were obtained.
>
> ### Blocksworld
>  Model               | 2-step | 4-step | 6-step | 8-step | 10-step | 12-step
> -|-|-|-|-|-|-
>  Llama-2-13B+RAP     | 0.44   | 0.42   | 0.38   | 0.11   | 0.00    | 0.00
>  Llama-2-13B+CoT     | 0.51   | 0.63   | 0.39   | 0.29   | 0.07    | 0.00
>  Llama-2-13B+RoT     | 0.49   | 0.70   | 0.30   | 0.07   | 0.00    | 0.00
>  Llama-2-13B+CRE     | 0.95   | 0.82   | 0.74   | 0.25   | 0.07    | 0.00
>  Llama-2-13B+CreDes  | -      | -      | -      | 0.65   | 0.49    | 0.37
>
> ### GSM8K
>  Model      | RAP  | RoT  | CoT  | CRE
> -|-|-|-|-
>  Llama-2-13B| 0.50 | 0.57 | 0.49 | 0.93
>
> ### Hanoi Tower
>  Model              | 3-step | 5-step | 7-step | 9-step | 11-step | 13-step
> -|-|-|-|-|-|-
>  Llama-2-13B+RAP    | 0.30   | 0.20   | 0.12   | 0.00   | -       | -
>  Llama-2-13B+CoT    | 0.33   | 0.24   | 0.09   | 0.03   | 0.00    | 0.00
>  Llama-2-13B+RoT    | 0.44   | 0.30   | 0.12   | 0.03   | -       | -
>  Llama-2-13B+CRE    | 0.42   | 0.38   | 0.27   | 0.10   | 0.01    | 0.00
>  Llama-2-13B+CreDes | -      | -      | -      | 0.34   | 0.15    | 0.07
>
> From the results, there is not much difference between the experimental results under 13B and 7B, and we believe that the difference can be regarded as a random error generated by different training. From the performance comparison between the 70B model and the 7B model under the RAP method, the performance of the 70B model will be relatively improved. However, considering inference speed, the 70B model is much slower than the 7B, and it needs to be loaded with a certain amount of quantization, and the performance loss is equally present.
> ## 2. Lack of Error Analysis:
> We agree on the need for detailed error analysis. Errors vary with different reasoning structures. (Using a Blocksworld 4-steps question as an example):
>
> *Considering the Characters Limitation,  please see the Official Comment.*
>
> Our method is closest to the expected output, with RAP errors including correct paths with wrong outcomes and vice versa, while RoT and CoT errors often occur when attention at the reasoning chain's end is ineffective.
> ## 3.Dataset Validity and Construction:
> The Hanoi Tower dataset is more complex than Blocksworld, involving a judgment on stacking order. Errors arise if the stacking order is violated, making the task harder. The dataset's size matches Blocksworld, with all steps being odd numbers based on the minimum steps required.
> ## 4.Computational Efficiency and Scalability:
> The 7B models fit within a single A100 GPU which is mentioned in paper. The 13B models have similar time requirements, as quantization isn't needed. However, 70B models experience significant speed drops, likely due to quantization and their size. Despite these challenges, our work shows potential for expansion due to its time efficiency.
> ## 5.Generalization to Less Structured Tasks:
> Our work focuses on structured tasks to validate the CreDes framework. We are designing experiments for open-world scenarios like Minecraft, referencing works such as Jarvis-1 and interactive planning with large language models. These experiments will help generalize our approach to open-world tasks and advance towards embodied intelligence.
>
> [1]. Wang Z, Cai S, Liu A, et al. Jarvis-1: Open-world multi-task agents with memory-augmented multimodal language models[J]. arXiv preprint arXiv:2311.05997, 2023.
>
> [2]. Wang Z, Cai S, Chen G, et al. Describe, explain, plan and select: Interactive planning with large language models enables open-world multi-task agents[J]. arXiv preprint arXiv:2302.01560, 2023.
> ## 6.Reporting Error Bars and Statistical Significance:
> We apologize for not including error bars and statistical significance. The revised paper will address this to validate our results. The current data suggests the error is insignificant.
> # Questions:
> 1.Please refer to Weakness 1 for the description.
>
> 2.Validation method: Similar to Blocksworld. Outputs are correct if they match expectations; otherwise, tested using PDDL. Violations of stacking order rules mark outputs as wrong, Hanoi Tower is stricter than Blocksworld.
>
> 3.Training speed difference is minimal between our model and the baseline, with the main difference in inference speed. RAP uses Monte Carlo search requiring multiple inferences, while CoT involves chained inferences. Our approach outputs answers directly, reducing search expansions compared to RAP and CoT.
>
> 4.GSM8K experiments address less structured tasks, with open-world reasoning tasks planned for future work. Refer to the Weaknesses section for details.
>
> 5.The inference process has less memory overhead, and our DES search method unfolds bidirectionally, allowing simultaneous head and tail tree searches, as shown in Fig.1. Our approach outputs short step sets (under 6-7 steps for 7B models) directly from the model which is different from RAP.
>
> We hope these clarifications address your concerns and improve the overall understanding of our work. We are committed to enhancing the paper based on your valuable feedback. Thank you for your time and consideration.
>
> ***We respect the rules of rebuttal, but there is a part of the question you want to know that we can't write in 6,000 characters, so we have created a separate official comments section for that, and would appreciate it if you would take it into consideration.***

---

> ### Author Response · Authors · 2024-08-07
> **Error analysis provided to Reviewer nWAC**
>
> To Reviewer nWAC:
>
> Due to the input limit of 6000 characters in the Rebuttal input box, we have no choice but to adopt this approach to improve our statement. This message may be released before the start of the discussion, and we apologize for any inconvenience caused. You should be able to see the complete statement after the debate begins.
>
> # Error analysis provided to Reviewer nWAC:
>
> Initial State:
> ```
> The blue block is clear, the orange block is clear, the hand is empty, the blue block is on top of the yellow block, the orange block is on top of the red block, the red block is on the table and the yellow block is on the table.
> ```
> Goal State:
> ```
> The orange block is on top of the yellow block.
> ```
> Expected output:
> ```
> <unstack the blue block from on top of the yellow block>
> <put down the blue block>
> <pick up the orange block>
> <stack the orange block on top of the yellow block>
> ```
> Actual output:
> ### RAP:
> A structure that unfolds in a tree shape:
>
> First Layer:
> ```
> <unstack the blue block from on top of the yellow block>
> ```
> Second Layer:
> ```
> <put down the blue block>
> or  <put down the yellow block> (hallucination)
> ```
> Third Layer:
> ```
> <unstack the orange block from on top of the red block> (hallucination)
> or  <unstack the orange block from on top of the yellow block> (hallucination)
> or  <unstack the orange block from on top of the blue block> (hallucination)
> ```
> Fourth Layer:
> ```
> <stack the orange block on top of the yellow block>
> or  <stack the orange block on top of the blue block> (pruned)
> ```
> ### CoT:
> The logic of CoT reasoning output is to solve complex problems by step-by-step reasoning and refining intermediate steps, ensuring the accuracy and reliability of the final answer.
> First Input: Initial State
>
> First Output:
> ```
> <unstack the blue block from on top of the yellow block>
> <put down the blue block>
> ```
> Second Input: Initial State + First Output:
>
> Second Output:
> ```
> <pick up the blue block> (hallucination)
> <stack the orange block on top of the blue block>
> ```
> ### CRE(Ours):
> Model one-time output of the whole process:
> ```
> <unstack the blue block>
> <put down the blue block>
> <pick up the orange block>
> <stack the orange block>
> ```
> It should be clarified that CRE's mistake lies in the possibility of incomplete answers as mentioned above.
>
> ***The above appendix will be included in our revised version of the paper.***

---

> > ### Comment · Reviewer_nWAC · 2024-08-10
> > **Reply to the rebuttal**
> >
> > Thank you for your detailed response; I am less concerned with the model size and error analysis and willing to increase my score a bit, but I am still unconvinced on the generalization of the proposed method on addressing the long-range causal reasoning problems without more experimental comparison and theoretical analysis as pointed out by other reviewers.

---

> > > ### Author Response · Authors · 2024-08-13
> > >
> > > Dear Reviewer nWAC,
> > >
> > > We would like to express our sincere gratitude for your previous responses and engagement with our paper. We noticed that you raised some further questions after our initial rebuttal, and we have since provided additional responses to address your concerns.
> > >
> > > We wanted to check in to see if our latest responses have resolved your queries, and to ask if there are any remaining questions or issues that you would like us to address. We apologize for reaching out so directly, but with the discussion period nearing its end, we will soon be unable to participate in further discussion.
> > >
> > > Once again, thank you very much for your time and thoughtful feedback. We greatly appreciate your support and look forward to hearing from you.
> > >
> > > Best regards,
> > >
> > > Authors

---

> > > > ### Comment · Reviewer_nWAC · 2024-08-13
> > > >
> > > > Thank you for your detailed response. This paper shows promise and could find its audience, but there's room for improvement in writing style and presentation. The final draft should include the detailed theoretical analysis you've provided. I strongly recommend offering more comprehensive details on implementation, particularly in light of the high efficiency and performance scores achieved on those datasets with CREDES performance. I'm maintaining my current score.

---

> ### Author Response · Authors · 2024-08-11
>
> Dear Reviewer nWAC,
>
> Thank you for your feedback and your willingness to reconsider your evaluation of our work. We appreciate your concerns regarding the generalization of our proposed method, particularly in relation to addressing long-range causal reasoning problems.
>
> # 1. Regarding Experimental Comparison:
> Many reasoning tasks or long-range sequence decomposition tasks in real life fundamentally follow the same paradigm as our research work. We understand that you may still have concerns about the real-world applicability of our approach, so we would like to provide a few scenarios to help clarify it.
>
> For instance, in the scheduling of port container stacking or the arrangement of goods in warehouse areas, there are already mature algorithms in the logistics field. However, our work attempts to leverage Large Language Models (LLMs) to enhance reasoning in these contexts. The goal is to bridge the communication gap between human operators and algorithm engineers by using LLMs to facilitate clearer and more effective interactions. We hypothesize that LLMs can receive and understand human instructions, adjusting their actions accordingly, which would improve collaboration between humans and algorithms.
>
> While we haven't yet validated our approach with real-world data, the essence of many real-world reasoning or long-range sorting tasks closely aligns with the experimental paradigm we employed in our paper.
>
> To further illustrate this, consider the example of warehouse item arrangement. This task might involve organizing items based on criteria like size, weight, or frequency of access. Although it may initially seem like a complex, monolithic task, it can actually be decomposed into a series of smaller, more manageable sub-tasks. For instance, the first sub-task could involve categorizing items by size, followed by arranging them within sections based on weight, and finally, placing them in specific locations depending on access frequency. Each sub-task is interdependent, with the completion of one informing the next, thereby creating a continuous sequence of actions that leads to the overall goal.
>
> We have observed that many related works do not establish a strong connection with real-world scenarios, and our experimental scope is similar to theirs. Specifically, other works have adopted similar test scenario and dataset, which strengthens our confidence that our experiments are robust enough to validate our ideas. For example, in addition to the baseline papers we cited, the following three papers also used the blocks world scenario to validate the performance of task decomposition. This demonstrates that the examples or experiments provided in our paper are highly effective for validating our reasoning capabilities and are indeed verifiable.
>
> In fact, Blocksworld is recognized, and we have omitted three papers from our baseline. The following are relevant papers based on Blocksworld validation for effectiveness:
>
> ## In subsequent work:
> [1] Yu F, Jiang L, Kang H, et al. Flow of Reasoning: Efficient Training of LLM Policy with Divergent Thinking.
>
> The experimental results (based on LLaMA3 8B, which was released after the submission of our paper) are as follows:
>
>  Model              | 2-step | 4-step | 6-step
> -|-|-|-
>  Llama-3-8B    | 100.00   | 97.62   | 71.71
>
> [2] Liu Z, Hu H, Zhang S, et al. Reason for Future, Act for Now: A Principled Architecture for Autonomous LLM Agents.
>
> The experimental results are presented in Fig.8 of their paper. Their baseline model, LLaMA-33B, shows performance similar to ours, but it struggles with 8-step, 10-step, and 12-step scenarios.
>
> ## Concurrent work with ours:
>
> [1] Zhang S, Zheng S, Ke S, et al. How Can LLM Guide RL? A Value-Based Approach.
>
> We noticed this work during the preparation of our paper. Due to their reliance on the OpenAI API for open-source code, we did not include it as one of our baseline comparison methods. However, they also compared our baseline method, RAP, as shown in Fig.5. Our Step-6 accuracy is higher compared to theirs.
>
> Additionally, if you have any real-world datasets or validation scenarios, we would greatly appreciate it if you could share them with us. Access to such datasets or scenarios would allow us to further validate our work and ensure its practical applicability in real-world contexts.
>
> # 2. Regarding Theoretical Analysis:
> We would like to clarify which aspects of the theoretical analysis you would like us to elaborate on. We acknowledge that due to word limitations, the theoretical sections addressed to other reviewers were brief. If you have specific concerns, and if time permits, we will do our best to provide a complete explanation. In the revised version of the paper, we plan to enhance the theoretical analysis in the appendix, incorporating feedback from all reviewers, and we welcome any further suggestions you may have.
>
>
>
> Thank you for your time and consideration.
>
> Authors

---

> ### Author Response · Authors · 2024-08-12
>
> Dear Reviewer nWAC,
>
> Thank you for your valuable feedback on our manuscript. The theoretical analysis you inquired about has been addressed in detail in our response to Reviewer BKzM's comments. We recommend that you refer to our reply to Reviewer BKzM for the relevant information.
>
> https://openreview.net/forum?id=azkuhJBZXi&noteId=cbDKQGUyra
>
> https://openreview.net/forum?id=azkuhJBZXi&noteId=RTQb3tqD03
>
> We appreciate your support of our work, and please feel free to reach out if you have any further questions.
>
> Best regards,
>
> Authors

---

> ### Author Response · Authors · 2024-08-14
>
> Dear Reviewer nWAC,
>
> Thank you for your continued engagement and for your constructive feedback on our paper. We appreciate your recognition of the promise in our work and your thoughtful suggestions for improvement.
>
> We will certainly take your recommendations to heart, especially regarding the need to refine the writing style and presentation. We also acknowledge the importance of including the detailed theoretical analysis and more comprehensive implementation details, particularly in light of the high efficiency and performance scores achieved on the datasets with CREDES performance. We will make these aspects a priority in our final draft to enhance the clarity and impact of our paper.
>
> Your insights have been invaluable in guiding us toward a stronger submission, and we are committed to making the necessary revisions.
>
> Thank you once again for your time and consideration.
>
> Best regards,
>
> Authors

---

### Author Response · Authors · 2024-08-07
**To ACs and Reviewers**

Dear AC and Reviewers,

We would like to express our sincere gratitude for the time and effort you have dedicated to the evaluation of our work. Your valuable insights and expertise are deeply appreciated.

In response to your review comments, we have prepared detailed answers to address the concerns and inquiries you have regarding our paper. Should there be any unresolved issues, or should you need further clarification or have additional questions, please do not hesitate to let us know. We stand ready to provide any information or clarification that may assist you in your review.

We have actually written a more informative and easy-to-understand Rebuttal, but since we need to comply with the 6,000 character limit on Rebuttal submissions, we have had to simplify the presentation, and we apologize for any confusion this may have caused you. We would be happy to receive further questions from you so that we can present it to you more fully.

Thank you once again for your invaluable time and consideration. We eagerly look forward to hearing from you, and we hope to have the opportunity to respond to any feedback or questions you may have.

With warm regards,

Authors

---

### Comment · Area_Chair_jBaa · 2024-08-10

Dear Reviewers,

The authors have submitted the point-to-point response.
Could you all read the rebuttal and see if it addresses the main issues raised by your review?
Are there any unaddressed/unresolved issues?

AC

---

### Author Response · Authors · 2024-08-14

Dear PC, SAC, AC, and Reviewers,

We express our deepest gratitude for the time and effort each of you has dedicated to reviewing and discussing our submission. The valuable feedback we have received has been instrumental in refining our work and strengthening its contributions to the field.
The following is a brief summary of the discussion process:

# Model Size and Scalability
Reviewer nWAC raised concerns about the limitations of the model sizes used in our experiments and the scalability of our approach. Reviewer yRix also highlighted the need for further demonstration of generalization and scalability, noting that the problems studied tend to be more structured, and they also questioned the performance scalability as the model parameters increase.

* Our Response:
(a) During the rebuttal period, we supplemented our experiments with results from the LLaMA-2-13B model, constrained by our hardware capabilities. We also pointed out that the original paper provided experimental results from the Mixtral-8x7B model, which is larger than the 7B model. The results showed no significant difference between the 13B and 7B models, and this difference can be considered a random error due to different training instances. Comparing the performance of the 70B and 7B models under the baseline RAP method, the 70B model shows relative improvement. However, regarding inference speed, the 70B model is much slower than the 7B model and requires a certain level of quantization, which also leads to performance loss. The above experimental results will be included in the main body of the revised paper.\
(b) We addressed the concerns about problem structure and scalability by explaining that our ongoing work explores more unstructured, complex scenarios using the Minecraft simulator, aiming to advance embodied intelligence. We also noted that in similar concurrent and subsequent work, our results show advantages, confirming the effectiveness and verifiability of our experiments. Relevant references are provided in the rebuttal for Reviewer nWAC.
# Theoretical Analysis
Reviewer Y4qM and Reviewer BKzM raised several detailed questions about our methodology and theoretical analysis, such as why we use |ATE| as part of the loss function, the relationship between the strength of causal relationships and the size of ATE, and the unclear statements regarding our assumptions. Reviewer nWAC also pointed out similar issues in their follow-up questions.

* Our Response:
We are grateful for these questions, as they helped us fully recognize the shortcomings in our writing. We have provided a more detailed theoretical analysis:\
https://openreview.net/forum?id=azkuhJBZXi¬eId=cbDKQGUyra \
https://openreview.net/forum?id=azkuhJBZXi¬eId=RTQb3tqD03 \
In these responses, we further clarified the unclear and imprecise aspects of our paper, and we will provide a comprehensive explanation of our theoretical analysis in the revised version, correcting and refining any unclear parts. We want to sincerely thank Reviewer Y4qM, Reviewer BKzM, and Reviewer nWAC for their invaluable feedback.

# Error Analysis and Statistical Significance
Reviewer nWAC noted our initial submission's lack of error analysis and statistical significance.
* Our Response:
We apologize for the omission in the initial draft. We provided an example of error analysis during the rebuttal period, and in the appendix of the revised paper, we will include more examples. Additionally, we will add error bars to all experimental result data in the revised paper.

# Writing Style, Typos, and Presentation Quality
Reviewer nWAC, Reviewer yRix, Reviewer Y4qM, and Reviewer BKzM provided feedback on various issues related to writing style, typos, clarity of diagrams, and overall presentation quality.
* Our Response:
We are very grateful to the reviewers for their patience in reading the manuscript and providing specific suggestions. We have carefully recorded all the revision suggestions and will address each of them in the revised paper. We especially noted the requests from Reviewer BKzM and Reviewer nWAC for improvements in theoretical analysis, and we will make a concerted effort to incorporate these into the additional pages of the main text as fully as possible.

# Conclusion
After the Rebuttal and Discussion phases, we are pleased to see that we have received support from multiple reviewers:

Reviewer nWAC: This paper shows promise and could find its audience, but there's room for improvement in writing style and presentation.

Reviewer Y4qM: I have more confidence on the soundness of the paper after reading your responses.

Reviewer yRix: I would like to keep my score and believe that this paper is lightly above the acceptance bar.

Reviewer BKzM: Thanks for the detailed responses, which have solved my major concerns. I suggest the authors continue to improve this submission.

Finally, we would like to thank all the reviewers and AC for supporting our paper.

Best Wishes,

Authors

---

### Author Response · Authors · 2024-08-14
**Statement on Reviewer MmUh's Relevant Opinions**

Dear All,

We extend our sincere gratitude for the opportunity to present our work and for the constructive feedback we have received from the reviewers. The review process has been invaluable in helping us refine and improve our submission. However, we would like to respectfully bring to your attention a few concerns regarding the feedback provided by Reviewer MmUh, particularly about their engagement during the rebuttal and discussion phases.

(a) The reviewer mentioned: "The authors just apply the concept of ATE to the Chain-of-Thought (CoT) without thorough analysis."

This assessment may not fully reflect the distinctions between our approach and the CoT method. Our approach is significantly different, especially in the form of outputs, as outlined in our paper and highlighted in the error analysis provided to Reviewer nWAC. The logic behind our reasoning and the specific inputs to the LLM, including elements like prompts and instructions, differ markedly from those used in the CoT method.

(b)The reviewer mentioned: "But this approach's validity hinges on the assumption that LLMs can perfectly represent the real-world environment." and "In other words, since there is no alignment with the causal relationships in real-world scenarios, the implementation of this method does not prove that the reasoning is causally sound."

This may need to be clarified. Our study focuses on widely recognized datasets like Blocksworld and GSM8K, as well as our custom Hanoi Tower dataset. Although examples such as Blocks World and Tower of Hanoi may seem simple compared to the real world, they are an abstract expression of the actual complex tasks in the real world. They can reflect the central core of long-range task decomposition, so they still have strong persuasiveness when validated on these datasets. For instance, besides the baselines we cited, other studies, such as those by Yu et al., Liu et al., and Zhang et al., have also used Blocksworld to validate task decomposition performance. These examples show that our experimental setup is practical and verifiable for evaluating reasoning capabilities. We welcome real-world datasets or validation scenarios supporting our method's applicability in practical contexts.

(c) This reviewer has not participated in the discussion. We and AC have repeatedly reminded the Reviewer MmUh but have not received a response.

We have noticed that Reviewer MmUh has yet to engage in the discussion phase, which may suggest that the original review might not entirely reflect the current status of our work. We believe that active participation in the discussion is essential for a fair and comprehensive review process, and we hope this can be considered.

(d) Besides, considering that our paper introduces a straightforward and effective algorithm, presents comprehensive experiments, and raises no ethical concerns, it may not warrant a rejection.

The definition of a reject is:

>"3: Reject: For instance, a paper with technical flaws, weak evaluation, inadequate reproducibility, and incompletely addressed ethical considerations."

We hope our work's strengths will be considered in the final assessment.

We appreciate your understanding and consideration of these points. We are committed to improving our submission and ensuring that it meets the high standards expected by the community. Again, Thank you for your time and effort overseeing this review process.

Best Wishes,

Authors

---

### Decision · Program_Chairs · 2024-09-25

**Decision:**

Reject

**Comment:**

This paper introduces the CreDes framework, which enhances long-range reasoning in LLMs by combining causal relationship enhancement (Cre) and a bi-directional search method (Des). The reviewers recognized the potential of the method but raised concerns about scalability, generalization to less structured tasks, and clarity in the explanation of ATE usage. While the authors addressed most concerns through additional experiments and clarifications, some reviewers remain skeptical about the generalization of the method and the thoroughness of the theoretical analysis. The paper shows promise, but further improvements in presentation and more comprehensive validation are necessary. Overall, the recommendation leans towards rejection due to these unresolved issues.